# ROS produced by Dual oxidase regulate cell proliferation and haemocyte migration during leg regeneration in the cricket

Misa Okumura-Hirono[1,2], Tetsuya Bando[1,*], Yoshimasa Hamada[1,3], Motoo Araki[2] and Hideyo Ohuchi[1,*]

## ABSTRACT

Many animals regenerate lost body parts through several signalling pathways; however, the triggers that initiate regeneration remain unclear. In the present study, we focused on the role of reactive oxygen species (ROS) produced by the NADPH oxidase Dual oxidase (Duox) during cricket leg regeneration. The results showed that ROS levels were upregulated during leg regeneration and decreased by $Duox^{RNAi}$. In $Duox^{RNAi}$ nymphs, wound closure and scab formation were incomplete 2 days after amputation, and hypertrophy occurred in the distal region of the regenerating legs at 5 days after amputation. In addition, the hypertrophic phenotype was induced by $DuoxA^{RNAi}$ and NADPH oxidase inhibitor treatment. During hypertrophy, haemocytes, including plasmatocytes, oenocytoids and granulocytes, accumulated. Proliferation of haemocytes in regenerating legs was not increased by $Duox^{RNAi}$; however, haemocyte accumulation was regulated by the Spatzle (Spz) family molecules, which are Toll receptor ligands. As the exoskeleton of $Duox^{RNAi}$ nymphs was thinner than that of the control, excessive haemocyte accumulation can cause hypertrophy in $Duox^{RNAi}$ nymphs. Thus, Duox-derived ROS are involved in wound healing and haemocyte accumulation through the Spz/Toll signalling pathway during leg regeneration in crickets.

KEY WORDS: Regeneration, Reactive oxygen species (ROS), NADPH oxidase (Nox), Dual oxidase (Duox), Inflammation, *Gryllus bimaculatus*

## INTRODUCTION

Tissue regeneration is a phenomenon whereby animals regrow lost body parts or tissues. It is observed in several animal species, including planarians, cnidarians, crustaceans, insects, holothurians, teleost fish, and amphibians (Galliot and Ghila, 2010; Ricci and Srivastava, 2018). Tissue regeneration occurs in three steps: wound healing; cell proliferation and blastema formation; and regrowth

[1]Department of Cytology and Histology, Okayama University Graduate School of Medicine, Dentistry and Pharmaceutical Sciences, 2-5-1 Shikata-cho, Kita-ku, Okayama 700-8558, Japan. [2]Department of Urology, Okayama University Graduate School of Medicine, Dentistry and Pharmaceutical Sciences, 2-5-1 Shikata-cho, Kita-ku, Okayama 700-8558, Japan. [3]Division of Molecular Biology, Institute for Genome Research, Institute of Advanced Medical Sciences, Tokushima University, 3-18-15, Kuramoto-cho, Tokushima 770-8503, Japan.

*Authors for correspondence (tbando@cc.okayama-u.ac.jp; ohuchi-hideyo@okayama-u.ac.jp)

 T.B., 0000-0002-3038-7782; Y.H., 0000-0002-7143-5365; M.A., 0000-0002-3172-2482; H.O., 0000-0003-1961-606X

and repatterning (Pandita et al., 2024; Taghiyar et al., 2018; Zielins et al., 2016). The molecular mechanisms underlying regeneration have been studied in several animal species (Dolmatov, 2021; Fox et al., 2020; Ivankovic et al., 2019; Joven et al., 2019; Marques et al., 2019; Phipps et al., 2020; Vogg et al., 2019). Recent studies have focused on the relationship between tissue regeneration and the immune system (Godwin et al., 2013). Moreover, reactive oxygen species (ROS) released from damaged tissues have been identified as triggers for regeneration (Love et al., 2013) during wound healing and cell proliferation. ROS are produced by phagocytes and play important roles in the immune defence, including antibacterial activity (Moghadam et al., 2021). They are also known as signalling molecules and are associated with several diseases (Vermot et al., 2021).

ROS are produced in mitochondria primarily through the leakage of electrons from complexes I and III of the electron transport chain. Nicotinamide adenine dinucleotide phosphate (NADPH) oxidase, a membrane-bound enzyme complex, produces ROS, specifically superoxide, by transferring electrons from NADPH to molecular oxygen (Moghadam et al., 2021; Vermot et al., 2021). Superoxide is converted into hydrogen peroxide by superoxide dismutase. Hydrogen peroxide is broken down into water and oxygen by catalase and other antioxidant enzymes, such as glutathione peroxidase. Hydrogen peroxide is taken up by the cells and acts as a signalling molecule (Sies, 2017). In humans and mice, NADPH oxidases are encoded by seven genes, *NOX1-4* (*NOX2* is also known as *CYBB*), *NOX5* (human only) and *DUOX1-2*, whereas in insects there are two genes, *Nox5* and *Duox* (Dual oxidase) (Fig. S1; Moghadam et al., 2021). Except Nox5, these isoforms interact with p22/DuoxA, a Duox maturation factor, to deliver Duox to the cell membrane and regulate its enzymatic activity (Panday et al., 2015; Vullien et al., 2025).

When a body part is lost, ROS are immediately detected around the wound in animals with regenerative ability such as tail regeneration in *Xenopus* tadpoles (Love et al., 2013), head and tail regeneration in planarians (Jaenen et al., 2021; Pirotte et al., 2015), wing disc regeneration in *Drosophila* (Khan et al., 2017), head regeneration in *Hydra* (Wenger et al., 2014), caudal fin regeneration in zebrafish (Gauron et al., 2013) and tail regeneration in axolotls (Baddar et al., 2019; Carbonell et al., 2022). In these animals, increased ROS levels are detected during wound healing (Dunnill et al., 2017; Hunt et al., 2024), and either decrease after wound healing (Baddar et al., 2019; Carbonell et al., 2022; Gauron et al., 2013) or are sustained during blastema formation (Love et al., 2013). Apocynin, diphenyleneiodonium chloride (DPI) and VAS2870 are NADPH oxidase inhibitors that have been widely used in animals, which decrease ROS production and inhibit tissue regeneration (Baddar et al., 2019; Carbonell et al., 2022; Gauron et al., 2013; Love et al., 2013; Pirotte et al., 2015). During blastema formation and tissue regeneration, injury-induced ROS promote cell proliferation, which compensates for cell death (Gauron et al., 2013) through Wnt

activation (Love et al., 2013), ERK (Jaenen et al., 2021), JNK (Khan et al., 2017), Hippo/YAP1 (Carbonell et al., 2022), and other signalling pathways activating cell proliferation during regeneration. The insect NADPH oxidase genes *Nox5* and *Duox* are involved in the inflammatory response and lifespan of insects (Iatsenko et al., 2018). In *Drosophila*, the *moladietz* (*mol*) gene, which encodes DuoxA, promotes wing disc regeneration (Khan et al., 2017), indicating that ROS are also involved in tissue regeneration in other insects.

Two-spotted cricket (*Gryllus bimaculatus*) nymphs regenerate lost parts of their legs through several moults (Bando et al., 2018; Mito and Noji, 2008; Mito et al., 2002; Nakamura et al., 2008). Our previous study showed that depletion of insect macrophages, known as plasmatocytes, reduces cell proliferation and inhibits leg regeneration. *Gryllus* plasmatocytes express several Toll receptors and the scavenger receptor Crq/CD36. RNA interference (RNAi)-based knockdown of Toll receptors, the Toll receptor ligands Spatzle (Spz) and Spz2, and other Toll signalling molecules inhibits leg regeneration in crickets (Bando et al., 2022). Plasmatocytes migrate to the wound and release the insect cytokine Unpaired (Upd), which is a ligand in the Jak/STAT signalling pathway, to promote cell proliferation in regenerating legs (Bando et al., 2013), which is also regulated by Hippo signalling (Bando et al., 2009); this is consistent with observations in other regenerative animals (Hayashi et al., 2014; Moya and Halder, 2019; Riley et al., 2022; Zhong et al., 2024).

In this study, we aimed to investigate the role of ROS in cricket leg regeneration. Although the *Gryllus* genome contains both *Duox* and *Nox5* genes, only *Duox* expression levels decreased in plasmatocyte-depleted regenerating legs. RNAi against *Duox* inhibited wound healing in the regenerating legs 2 days post-amputation (dpa). Epidermal cell proliferation decreased, whereas haemocyte accumulation increased, causing hypertrophy in *Duox*^RNAi regenerating legs at 5 dpa. *spz* expression levels were upregulated in *Duox*^RNAi regenerating legs, and *spz*^RNAi and *spz2*^RNAi suppressed the hypertrophic phenotype induced by *Duox*^RNAi. Therefore, Duox-derived ROS are involved in wound healing and haemocyte accumulation through the Spz/Toll signalling pathway during leg regeneration in crickets.

## RESULTS
### Plasmatocytes promote epidermal cell proliferation
Our previous study showed that depletion of plasmatocytes by injecting clodronate liposomes (Clo-lipo) inhibited leg regeneration by downregulating cell proliferation in cricket nymphs (Bando et al., 2022). To analyse further the differences in tissue morphology and cell proliferation in Clo-lipo-treated regenerating legs, we amputated the tibia of the hindleg of third instar nymphs (Fig. 1A) and stained sections of the paraffin-embedded regenerating legs with Haematoxylin-Eosin (H-E). Next, we detected proliferating cells by incorporating 5-ethynyl-2′-deoxyuridine (EdU). At 2 dpa, the wound epidermis covered the wound site beneath the scab in the PBS-lipo-treated control nymphs (Fig. 1B), as revealed by H-E staining. In Clo-lipo-treated nymphs, scabs were formed; however, wound closure by the epidermis was incomplete (Fig. 1B). In PBS-lipo-treated regenerating legs, epidermal cells and cells along the trachea proliferated (Fig. 1C). However, in Clo-lipo-treated regenerated legs, the number of proliferating cells was significantly reduced (Fig. 1C,D). Previously, we reported that the expression levels of some Toll receptors, Toll ligands, components of Jak/STAT signalling pathway, and *cyclin* genes are downregulated in Clo-lipo-treated regenerating legs (Bando et al., 2022). Furthermore, we

hypothesised that the expression of the NADPH oxidases *Duox* and *Nox5* (Fig. 1E, Fig. S1) may be affected by the Clo-lipo treatment. *Duox* expression levels were significantly decreased to 30.9% in the Clo-lipo-treated regenerating legs (P<0.001, Student's t-test), whereas *Nox5* expression did not change (P>0.05, Student's t-test) (Fig. 1F). Because plasmatocyte depletion decreased *Duox* expression, we hypothesised that Duox could be involved in leg regeneration; therefore, we focused on the function of Duox during cricket leg regeneration.

To determine whether ROS production occurs during cricket leg regeneration, 4-hydroxy-2-nonenal (4-HNE) was used as a biomarker of lipid peroxidation targeted by oxidative stress (Tanaka et al., 1997; Toyokuni et al., 1995) because the ROS-detecting dyes Dihydroethidium, CellROX Reagents and BODIPY 581/591 C11 could not be used on the fixed regenerating legs of RNAi- or inhibitor-treated cricket nymphs. Immunofluorescence staining was performed using an anti-4-HNE antibody on sections of the regenerating legs at several time points, followed by staining with H-E to compare the tissue structures (Fig. 2, Fig. S2). In the unamputated legs, fluorescent signals were observed in the epidermis, muscles and cuticle (Fig. S2A,A′). Cuticular fluorescence was observed in the negative control stained with normal IgG (Fig. S2B,B′), indicating that the cuticle exhibited autofluorescence. The fluorescence intensities of anti-4-HNE antibody staining in the epidermis and muscles were similar (Fig. S2C). At 0 h post-amputation (hpa), strong signals were observed in the tarsal depressor muscle on the ventral side of the amputated leg (Fig. 2A,A′). In the distal region of the tibia, the tarsal depressor muscle is connected to the tarsus on the ventral side (Snodgrass, 1935). When the tibia was amputated, the connection of the muscle to the tarsus was disrupted, and the muscle may have shrunk in the tibia. The intense fluorescence observed in the distal part of the muscle may have been caused by shrinkage (Fig. 2A′). At 3 hpa, a thin eosinophilic clot was formed at the most distal portion of the amputated leg covering the amputation site, and haemocytes accumulated at the amputation site to form a scab (Fig. 2B). The ROS levels observed in the muscle were lower than those observed at 0 hpa (Fig. 2B′). At 48 hpa (2 dpa), wound closure occurred in the wound epidermis, and numerous haemocytes were observed between the scab and the wound epidermis (Fig. 2C). Increased ROS production was observed in the epidermis beneath the cuticles and the wound epidermis at 2 dpa (Fig. 2C′) compared with 0 and 3 hpa. Intense fluorescent signal was also observed in the accumulated haemocytes beneath the scab (Fig. 2C′). To identify the cell types producing ROS, we compared them with regenerating leg samples that had been injected with India ink and sectioned them. India ink-incorporating plasmatocytes were located along the trachea and accumulated in haemocytes beneath the scab (Fig. S3A), indicating that the ROS-producing cells were plasmatocytes (Fig. 2C′). At 96 hpa (4 dpa), the distal epidermis began to thicken (Fig. 2D) and at 144 hpa (6 dpa), the distal structure of the leg was reconstructed (Fig. 2E). The ROS levels in the epidermis at 96 and 144 hpa were lowered (Fig. 2D′,E′). Measurement of fluorescence intensities of the epidermis revealed that ROS production increased significantly at 48 hpa (2 dpa) (P<0.01, Tukey's test) during cricket leg regeneration (Fig. 2F).

ROS production temporarily increased at 48 hpa during regeneration (Fig. 2F), and *Duox* expression was downregulated in Clo-lipo-treated regeneration-defective legs (Fig. 1F), indicating that Duox can produce ROS during leg regeneration. We quantified the *Duox* changes and determined that *Duox* expression was high at 0 hpa and reduced to <20% during leg regeneration (Fig. S3B). Similar changes in expression were observed in the quantitative

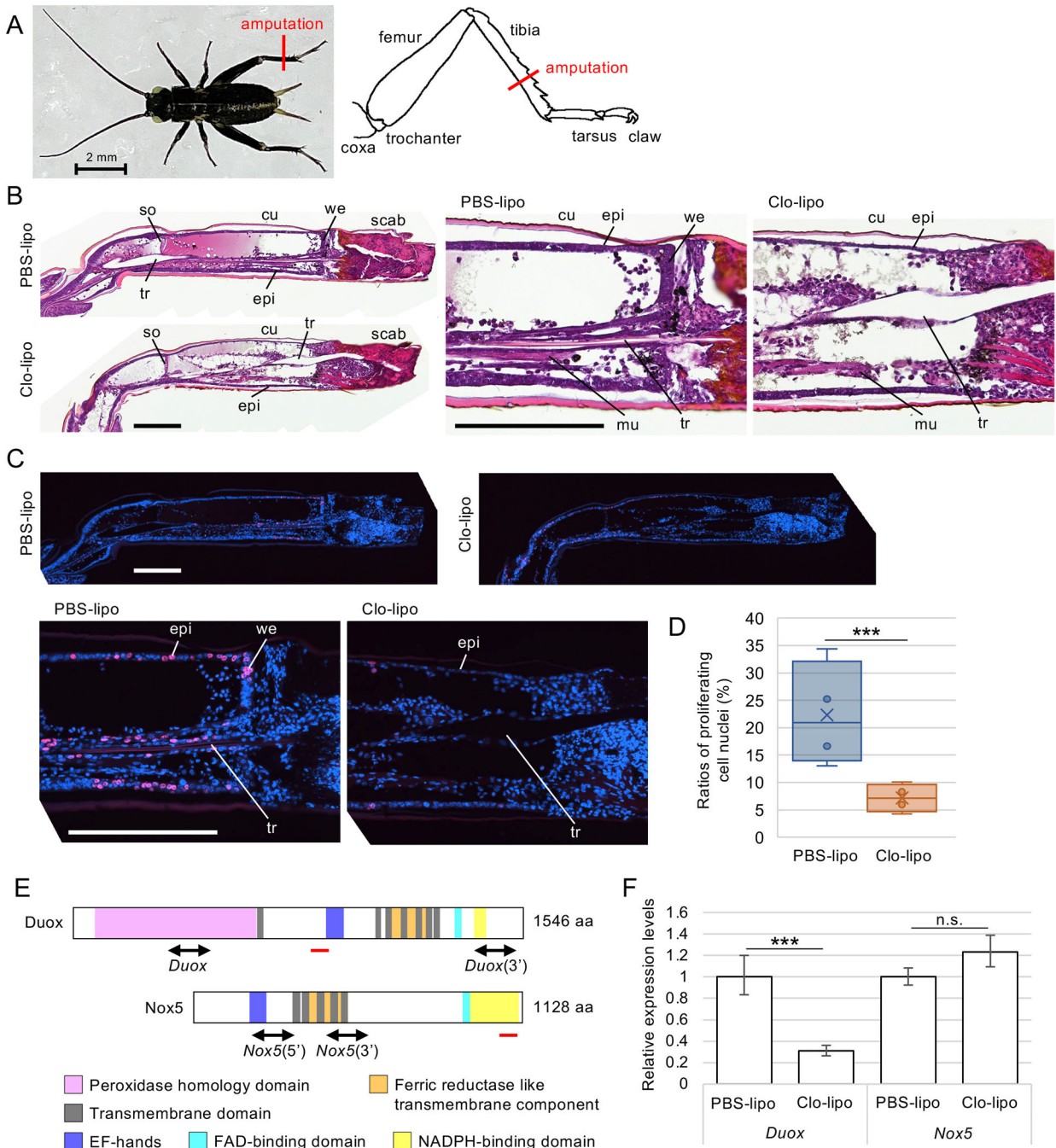

**Fig. 1. Wound healing, cell proliferation and gene expression in third instar nymphs at 2 dpa.** (A) Image of a two-spotted cricket *Gryllus bimaculatus* third-instar nymph and diagram of the leg, showing the site of amputation. The leg of the cricket is divided into six parts: coxa, trochanter, femur, tibia, tarsus and claw. To induce leg regeneration, we amputated the tibia. (B) H-E-stained sections of PBS-lipo- and Clo-lipo-treated regenerating legs at 2 dpa. Plasmatocytes were visualised by India ink incorporation. cu, cuticle; epi, epidermis; mu, muscle; so, subgenual organ; tr, trachea; we, wound epidermis. Images are representative of ten samples. (C) Fluorescence microscopy images of the EdU incorporation assay in PBS-lipo- and Clo-lipo-treated regenerating legs at 2 dpa. EdU-incorporating S-phase nuclei are shown in magenta and all nuclei (stained with Hoechst 33342) are shown in blue. Scale bars: 200 µm (B,C). (D) The ratios of proliferating cell nuclei of PBS-lipo- and Clo-lipo-treated regenerating legs are shown in box plots. ***$P<0.001$ (unpaired, two-tailed Student's *t*-test between PBS-lipo- and Clo-lipo-treated regenerating legs). *n*=4. (E) Schematic of *Gryllus* Duox and Nox5 proteins. Double-headed arrows and red lines indicate regions for dsRNA and qPCR amplicons, respectively. (F) Relative expression levels of *Duox* and *Nox5* transcripts in PBS-lipo- and Clo-lipo-treated regenerating legs at 2 dpa, as revealed by qPCR. The *y*-axis indicates normalised expression after PBS-lipo-treatment and relative expression levels after Clo-lipo-treatment. ***$P<0.001$ (unpaired, two-tailed Student's *t*-test between PBS-lipo- and Clo-lipo-treated regenerating legs); n.s., not significant. *n*=8.

PCR (qPCR) results for *paramyosin* (Fig. S3C), which encodes a thick filament protein of muscle cells, indicating that the ROS production observed in the muscle (Fig. 2A′) was elicited by Duox. We also quantified the expression of *DuoxA*, which regulates Duox activity, and determined that DuoxA expression was highest at 0 hpa, temporally high at 2 dpa, and reduced from 3 hpa to 6 dpa (Fig. S3D), indicating that DuoxA activated Duox to produce ROS at 2 dpa.

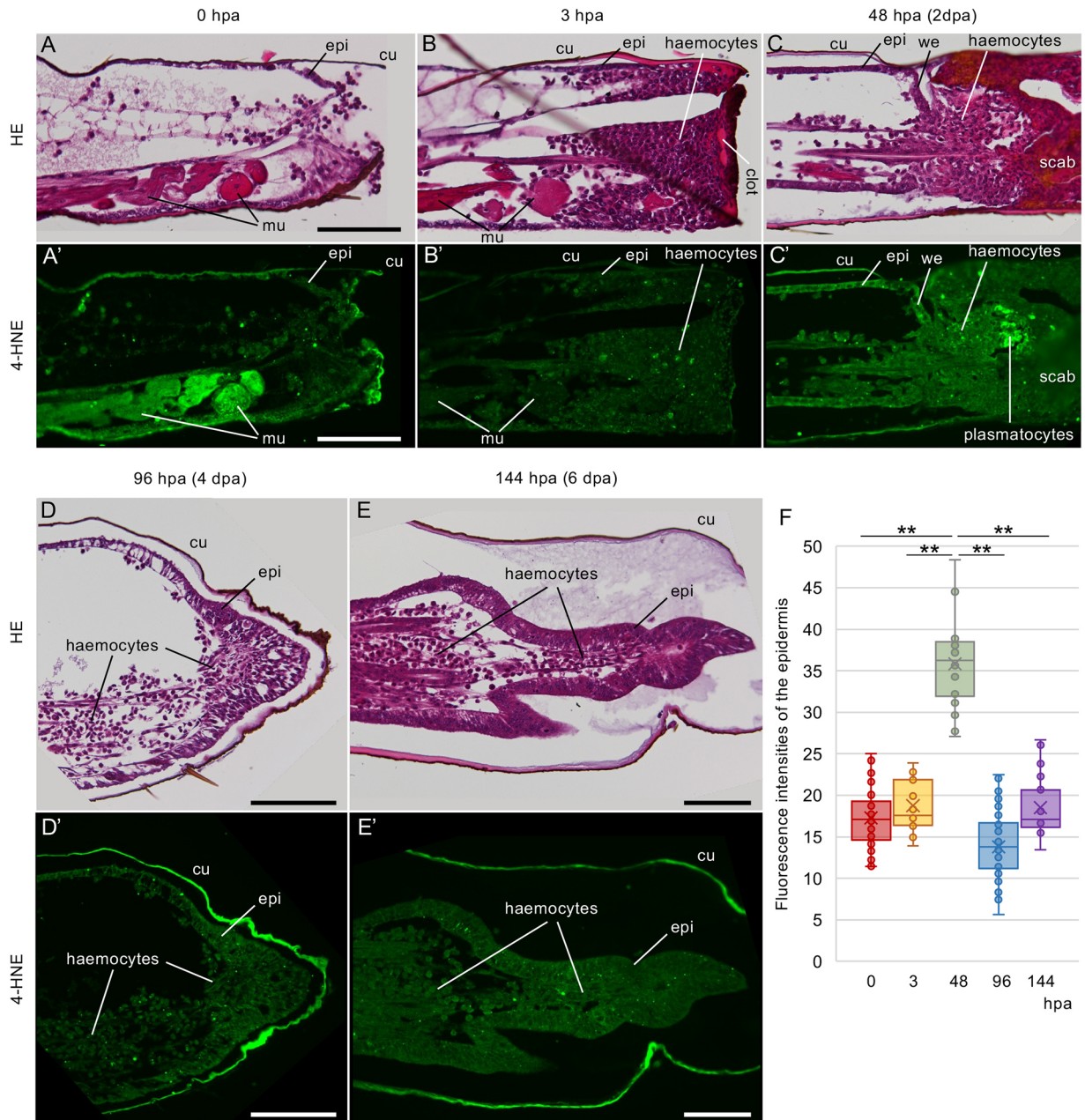

**Fig. 2. Morphological and ROS production changes during leg regeneration.** (A-E′) Temporal changes in morphology and ROS production in high-magnification images of the distal leg structures during regeneration. Bright-field images for histology (A-E) and respective fluorescence microscopy images for 4-HNE localisation (A′-E′) in regenerating legs at 0 (A,A′), 3 (B,B′), 48 (C,C′), 96 (D,D′) and 144 (E,E′) hpa are shown. cu, cuticle; epi, epidermis; mu, muscle; we, wound epidermis. Images are representative of three samples. Scale bars: 100 μm. (F) The fluorescence intensities of epidermis in each image were measured using ImageJ software and are shown in box plots. The fluorescence intensity at 48 hpa was significantly higher than those at other time points. **$P<0.01$ (Tukey–Kramer test). $n$=18.

## Duox is involved in scab formation and wound healing

We hypothesised that Duox may be involved in leg regeneration in crickets because of the induction of ROS production and changes in the expression levels of *Duox* and *DuoxA* during leg regeneration. Thus, we cloned 5′ and 3′ partial fragments of *Duox*, designated as *Duox* and *Duox(3′)*, respectively, and used these as DNA templates to synthesise the respective double-stranded RNA (dsRNA) for RNAi (Fig. 1E). We injected *Duox* dsRNA and amputated one-third of the distal position in the tibiae of the legs of third instar nymphs to induce leg regeneration (Fig. 1A). *Duox*[RNAi] and *Duox(3′)*[RNAi] decreased the relative amounts of endogenous *Duox* mRNA to

21.5% and 5.2%, respectively, at 2 dpa, compared with the RNAi against exogenous gene *DsRed*[RNAi] as control, as revealed by qPCR (Fig. 3A, Fig. S4A). At 2 dpa, the amputated sites of both *Duox*[RNAi] and control regenerating legs were covered by scabs, and no obvious differences were observed between the *Duox*[RNAi] legs and control legs using a dissecting microscope (Fig. 3B). In H-E-stained sections of *Duox*[RNAi] regenerating legs at 2 dpa, scab formation was incomplete, and wound closure was delayed compared to the control legs (Fig. 3B), similar to the phenotypes observed in the Clo-lipo-treated regenerating legs (Fig. 1B). In *Duox*[RNAi] regenerating legs, haemocytes localised between the wound

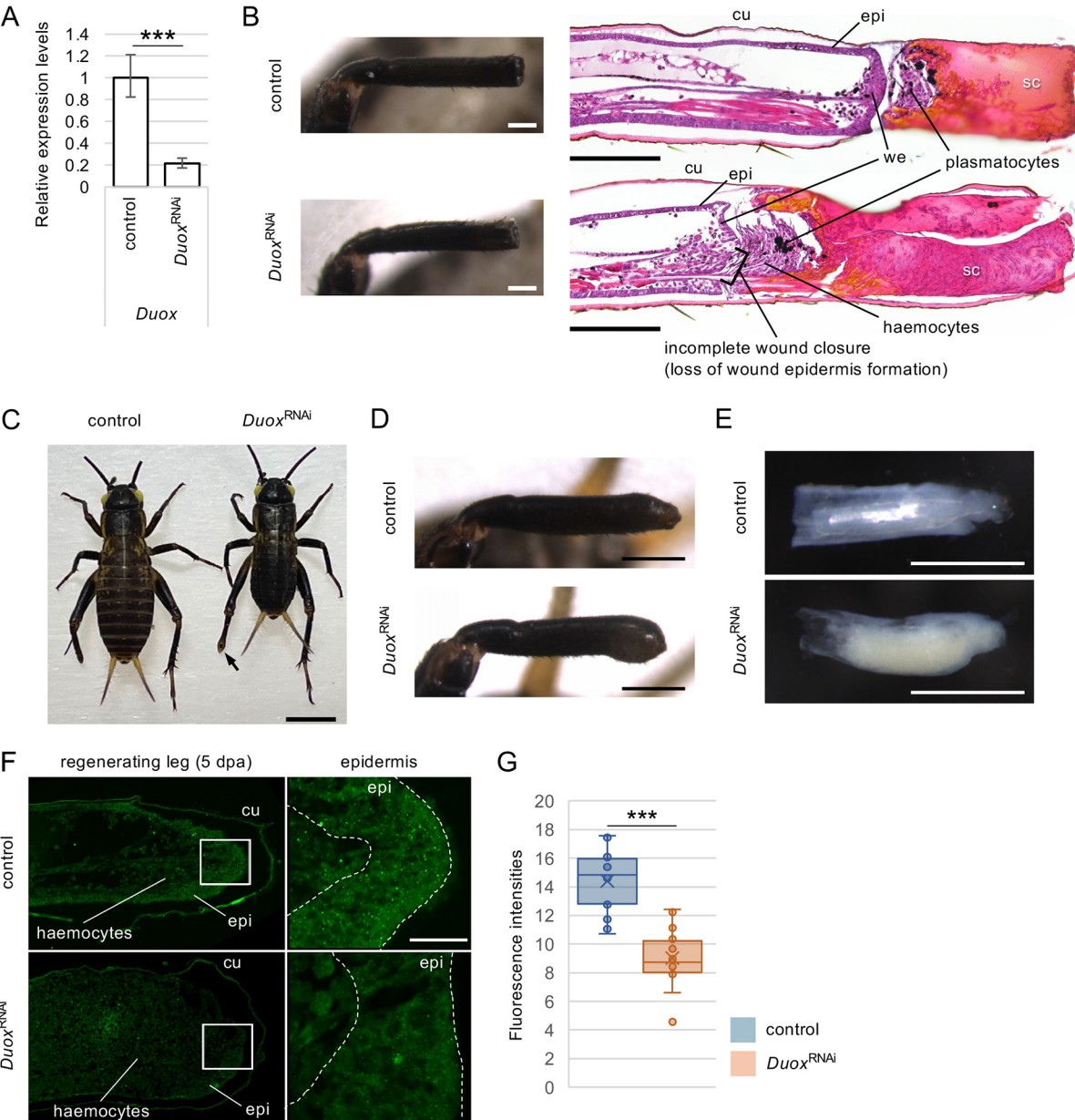

**Fig. 3. Phenotypes of *Duox*^RNAi nymphs at 2 and 5 dpa.** (A) Relative *Duox* transcript expression levels in control and *Duox*^RNAi regenerating legs at 2 dpa. ***$P<0.001$ (unpaired, two-tailed Student's *t*-test between control and *Duox*^RNAi crickets). *n*=8. (B) Typical morphology of regenerating legs in bright-field and H-E-stained sections of the control and *Duox*^RNAi regenerating legs at 2 dpa. Plasmatocytes were visualised by India ink incorporation. cu, cuticle; epi, epidermis; sc, scab; we, wound epidermis. (C-E) Typical morphology of whole nymphal bodies (C), regenerating legs (D) and exoskeleton-removed regenerating legs (E) of control and *Duox*^RNAi nymphs at 5 dpa in the fourth instar. (F) Fluorescence microscopy images of control and *Duox*^RNAi regenerating legs stained with anti-4-HNE antibodies to observe changes in ROS levels in the epidermis. Boxed regions are shown at higher-magnification on the right.. cu, cuticle; epi, epidermis. Images are representative of three samples. (G) Fluorescence intensities of epidermis measured using ImageJ software are shown in box plots. ***$P<0.001$ (unpaired, two-tailed Student's *t*-test between fluorescence intensities of epidermis and mesenchyme in control and *Duox*^RNAi). *n*=20. Scale bars: 200 μm (B); 2 mm (C); 500 μm (D,E); 20 μm (F).

epidermis and scabs increased compared to those in the control. Plasmatocytes, which were visualised by the incorporation of India ink, accumulated within the haemocytes.

### Duox is involved in leg regeneration and nymphal growth

In the fourth instar, *Duox*^RNAi nymphs showed hypertrophy of the regenerating legs (Fig. 3C,D). In addition, most *Duox*^RNAi nymphs were smaller than the control nymphs (Fig. 3C, Fig. S5A), and their moulting from the fourth to fifth instar was inhibited, causing lethality at the nymph stage after approximately 1 month of survival as the fourth instar. Because control nymphs sank when placed in a fixative solution but *Duox*^RNAi nymphs did not sink, we considered the possibility that the *Duox*^RNAi nymphs had abnormalities in their digestive tract, causing air to accumulate. The digestive system of *G. bimaculatus* is divided into the oesophagus, crop, proventriculus, caecum, and other organs along the head-to-tail axis (Woodring and Lorenz, 2007). In the foregut of *Duox*^RNAi nymphs, the crop was larger; however, the crop contained a small amount of undigested food or was empty, and the caecum was atrophied compared to that of control nymphs (Fig. S5B). In addition, notably less adipose

tissue was observed inside the exoskeleton in $Duox^{RNAi}$ nymphs than in controls (Fig. S5C). These results indicate that the nymphal lethal phenotype of $Duox^{RNAi}$ may be attributed to abnormalities in the digestive organs and defects in fat storage.

In the regenerating legs, hypertrophy of exoskeleton-removed regenerating legs of $Duox^{RNAi}$ and corresponding regions of control nymphs were observed at 5.5 dpa (Fig. 3E). In the control regenerating leg, the tracheas were observed inside the epidermal tissue, and the tibial spurs and tarsal segments were regenerated. In $Duox^{RNAi}$ regenerating legs, a population of whitish cells accumulated in the distal regions, a phenomenon that was not observed in control-regenerating legs. The accumulation of whitish cells may have caused hypertrophy (Fig. 3E). Tarsal structures were also regenerated on the distal side of the tibia in $Duox^{RNAi}$ regenerating legs.

To examine whether the hypertrophy and growth arrest phenotypes were caused by the reduction of $Duox$ expression, we examined the $Duox(3')^{RNAi}$ phenotype at 5 dpa. $Duox(3')^{RNAi}$ nymphs also showed hypertrophy in distal region of regenerating legs at the fourth instar (Figs S4A,B), and the nymphs had a smaller body size and did not moult to the fifth instar, similar to the phenotypes observed in $Duox^{RNAi}$ nymphs. The consistent phenotypes shown by RNAi against different regions of $Duox$ indicate that $Duox^{RNAi}$ caused these phenotypes (Fig. 3C,D, Fig. S4B,C).

$Duox^{RNAi}$ and $Duox(3')^{RNAi}$ showed similar hypertrophy phenotypes (Fig. S4C), indicating that ROS production by Duox is important for leg regeneration. Hence, we confirmed a reduction in ROS levels in $Duox^{RNAi}$ regenerating legs at 4 dpa, as revealed by anti-4-HNE antibody staining. In high-magnification images of the distal regions of regenerating legs, green fluorescent particles were distributed in the epidermis of the control; however, they decreased in the epidermis of $Duox^{RNAi}$ regenerating legs (Fig. 3F). To quantify the decrease in 4-HNE levels, the fluorescence

intensities were measured in the epidermis. The fluorescence intensity of the epidermis in the $Duox^{RNAi}$ regenerating legs decreased significantly to 62.6% ($P<0.001$, Student's $t$-test) compared to the control (Fig. 3G), indicating that the hypertrophy observed in $Duox^{RNAi}$ regenerating legs is correlated with the reduction of ROS.

Duox-mediated ROS production is regulated by DuoxA (DuoxMF) 1/2 (Fig. 4A), which forms a complex with Duox in the cell membrane (Khan et al., 2017). DuoxA is an evolutionarily conserved molecule; the arthropod genome contains a single $DuoxA$ gene, although mammals and some other phyla have two $DuoxA$ genes (Fig. S6A) (Vullien et al., 2025). Therefore, we compared the phenotypes of $DuoxA^{RNAi}$ and $Duox^{RNAi}$. $DuoxA^{RNAi}$ significantly decreased endogenous $DuoxA$ transcripts to 5% ($P<0.001$, Student's $t$-test) (Fig. 4A,B), and $DuoxA^{RNAi}$ nymphs showed hypertrophy in the regenerating legs at the fourth instar stage (Fig. 4C), similar to the $Duox^{RNAi}$ phenotype (Fig. 3C,D, Fig. S4B). To examine whether $DuoxA^{RNAi}$ synergistically affects $Duox^{RNAi}$, we performed dual RNAi against $Duox$ and $DuoxA$. $Duox/DuoxA^{RNAi}$ also led to hypertrophy in regenerating legs, and some nymphs showed marked hypertrophy and a shortened regenerating leg phenotype compared to the $Duox^{RNAi}$ phenotype (Fig. 4D).

Next, we examined the effect of ROS reduction by treatment with DPI, an NADPH oxidase inhibitor that has been used in other animal models of regeneration (Baddar et al., 2019; Gauron et al., 2013; Love et al., 2013; Pirotte et al., 2015). We injected DPI at several concentrations, 1, 10 or 50 µM, at the third instar and amputated the legs at the tibia. At 5 dpa, hypertrophy was observed in the distal regions of the regenerating legs at all DPI concentrations (Fig. 4E), and no other phenotypes were observed.

$Duox^{RNAi}$, $DuoxA^{RNAi}$ and DPI treatments resulted in similar hypertrophic phenotypes in regenerating legs, indicating that ROS

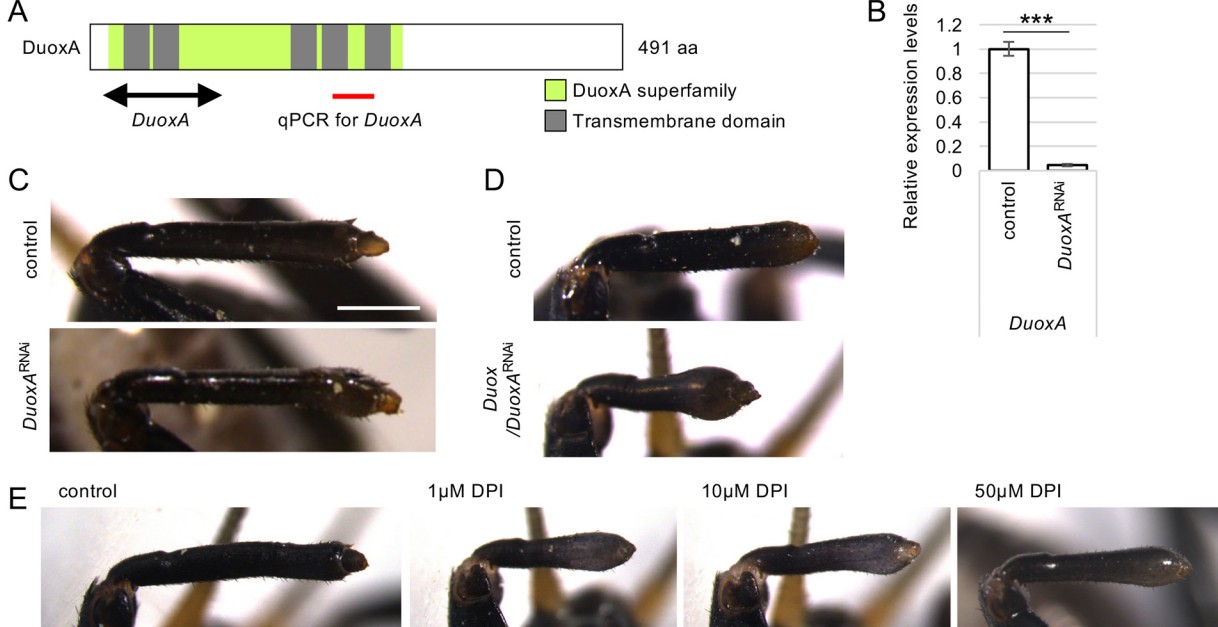

**Fig. 4. Typical phenotypes of $DuoxA^{RNAi}$ and DPI-treated nymphs.** (A) Domain structures of *Gryllus* homologue of DuoxA. Green box and grey boxes indicate DuoxA superfamily domain and transmembrane domains, respectively. Double-headed arrow and red line indicate regions of dsRNA and qPCR amplicon, respectively. (B) Relative $DuoxA$ transcript expression levels in regenerating legs of the control and $DuoxA^{RNAi}$ nymphs at 2 dpa. ***$P<0.001$ (unpaired, two-tailed Student's $t$-test between control and $DuoxA^{RNAi}$). $n=8$. (C) Typical phenotypes of regenerating legs of the control and $DuoxA^{RNAi}$ nymphs at 5 dpa. Scale bar: 500 µm. (D) Typical phenotypes of regenerating legs of the control nymph and dual RNAi nymph against $Duox/DuoxA$ at 5 dpa. (E) Typical phenotypes of regenerating legs of the control reagent-treated and 50 µM, 10 µM or 1 µM DPI-treated nymphs at 5 dpa. Images are representative of 24 samples.

production from Duox is important for proper leg regeneration. In contrast to $Duox^{RNAi}$, the $Nox5^{RNAi}$ group did not show any obvious morphological changes in the regenerating legs than the control, although endogenous $Nox5$ transcripts were significantly decreased by RNAi (Fig. S6B,C).

Hydrogen peroxide produced by Duox is broken down by catalase. To test the importance of ROS during leg regeneration, we performed RNAi against *Catalase* (Fig. S7A). $Cat(5')^{RNAi}$ and $Cat(3')^{RNAi}$ decreased endogenous *Catalase* expression (Fig. S7B) and led to defects in segmentation of the regenerated tarsus in the fifth instar (Fig. S7C), but no hypertrophic phenotype during the fourth instar. We also performed RNAi against *Nrf2* (Fig. S7D), which encodes a transcription factor responsible for oxidative stress. $Nrf2(5')^{RNAi}$ and $Nrf2(3')^{RNAi}$ decreased endogenous *Nrf2* expression (Fig. S7E), and led to nymphal lethality caused by defects in moulting and melanisation (Fig. S7F), which may have been caused by the decreased expression of *pale*, which encodes a tyrosine hydroxylase. The regenerating legs of $Nrf2^{RNAi}$ nymphs did not show hypertrophy. We concluded that ROS regulation by Catalase and Nrf2 are involved in regenerated leg patterning and nymphal growth.

## Haemocytes accumulated in the regenerating legs cause hypertrophy

To analyse histologically the cell population of the whitish tissue in hypertrophy (Fig. 3E), we sectioned the regenerating legs at 6 dpa and stained them with H-E (Fig. 5A). Distal regions of the amputated positions, indicated with arrowheads in Fig. 5A, were regenerated, and the most distal epidermis of the control and $Duox^{RNAi}$ nymphs was thickened compared with the epidermis in the stump regions. In the control group, the lost part of the tarsus began to be reconstructed by the thickened epidermis, although reconstruction did not start in the $Duox^{RNAi}$ nymphs. The trachea extended from the amputated position to the epidermis. The muscular tissues on the ventral side were not connected to the regenerating tarsus. Mesenchymal cells were widely spread throughout the entire regenerating leg in the control group, but densely accumulated, especially in the hypertrophic region of the leg in the $Duox^{RNAi}$ nymph group (Fig. 5A). The insect epithelial tissue is protected by an exoskeleton, which is composed of three cuticle layers from the outside to the inside: the epicuticle, exocuticle and endocuticle (Nation, 2008). In high-magnification images of the cuticle layers, the exocuticle of $Duox^{RNAi}$ regeneration legs was thinner and paler in colour than that of control legs (Fig. S5D), indicating that the exoskeleton of $Duox^{RNAi}$ nymphs was softer than that of the control.

High-magnification images of three typical mesenchymal cells in the hypertrophic regions of regenerating legs are shown in Fig. 5B-B″. Because the spindle-shaped mesenchymal cells (Fig. 5B) were plasmatocytes (Bando et al., 2022), other mesenchymal cells may also be haemocytes (Cho and Cho, 2019). Most of the other mesenchymal cells in $Duox^{RNAi}$ regenerating legs were classified into two types: eosinophilic round-shaped cells with flat nuclei (Fig. 5B′) and basophilic round-shaped cells with round nuclei, including a single nucleolus (Fig. 5B″). To identify the haemocyte types, we quantified haemocyte marker gene expression. We prepared cDNAs from the hypertrophic regenerating legs of $Duox^{RNAi}$ nymphs and the regenerating legs of control nymphs at 5 dpa and used them for qPCR. In hypertrophic $Duox^{RNAi}$ regenerating legs, the expression of *Toll2-1*, *Toll2-2* and *Toll2-5*, which are plasmatocyte marker genes, was significantly increased ($P<0.001$, Student's *t*-test), as expected (Fig. 5C). In addition, the expression of the phenol oxidase genes *PO2* and *PO1-1*, which are marker genes of oenocytoids (Joseph,

2014), and *integrin PS3* (Zhang et al., 2014), which is a marker of granulocytes, was also significantly increased ($P<0.001$, Student's *t*-test) (Fig. 5C), indicating that the typical mesenchymal cells shown in Fig. 5B-B″ were haemocytes of plasmatocytes, oenocytoids and granulocytes, respectively.

## Duox is not involved in haematopoiesis

In our previous study, RNAi against *yorkie* (*yki*), which encodes a transcriptional co-activator that strongly promotes cell proliferation under Hippo pathway regulation (Cho et al., 2006), suppressed the enlargement of regenerating legs caused by RNAi against *fat*, *dachsous* or *warts* (Bando et al., 2009). Thus, we performed dual RNAi against *yki* and *Duox* to clarify whether cell proliferation regulated by Yki was involved in the hypertrophy caused by $Duox^{RNAi}$. Almost all $yki^{RNAi}$ crickets showed nymphal lethality during moulting from the third to the fourth instar; however, only a few grew to the fourth instar. The $yki^{RNAi}$ crickets did not exhibit hypertrophy (Fig. S8A). Dual RNAi against *Duox/yki* suppressed hypertrophy, indicating that Yki-dependent cell proliferation is involved in the hypertrophic phenotype of $Duox^{RNAi}$.

To elucidate whether hypertrophy in regenerating legs was caused by haemocyte proliferation, we analysed which cells were proliferating using the EdU incorporation assay. In the control regenerating legs at 4 dpa, haemocytes were scattered in the distal mesenchymal region of the regenerating leg, as revealed by Hoechst 33342 staining (Fig. 5D). In the high-magnification image of the distal region, EdU-incorporated proliferating cells were detected in 45.5±8.1% (mean±s.d.) of the epidermis (Fig. 5E); however, only a few were observed in the mesenchyme. Hoechst 33342 staining revealed that haemocytes accumulated in $Duox^{RNAi}$ regenerating legs, especially in the distal hypertrophic region (Fig. 5D). EdU-incorporated cells were significantly reduced to 2.2±1.2% in the epidermis (Fig. 5E), and the accumulated haemocytes did not proliferate (Fig. 5D), indicating that haemocyte proliferation did not increase in $Duox^{RNAi}$ regenerating legs.

Hypertrophy is not caused by haemocyte proliferation; however, whether it is related to increased haematopoiesis remains unclear. In *G. bimaculatus*, haemocytes were derived from two pairs of triangle-shaped lymph glands located on each side of the dorsal vessel in the second and third abdominal segments (Fig. S8B) (Grigorian and Hartenstein, 2013). We performed the EdU incorporation assay on the lymph glands. In the fourth instar, EdU-incorporated cells were distributed in the lymph glands of control and $Duox^{RNAi}$ nymphs (Fig. S8C). We counted the number of EdU-incorporated and total nuclei and calculated their ratios. The average numbers of EdU-incorporated nuclei were 176.0±52.7 and 156.5±18.8, and average total nuclei were 851.3±149.8 and 739.8±95.6 in the control and $Duox^{RNAi}$ nymphs, respectively (Fig. S8D). The ratios of EdU-incorporated nuclei per total nuclei were 20.6±5.0 and 21.4±2.9 in the control and $Duox^{RNAi}$ nymphs (Fig. S8D), respectively. The ratio of EdU-incorporated nuclei in $Duox^{RNAi}$ nymphs to that in control nymphs was not significantly different ($P>0.05$, Student's *t*-test), indicating that the reduction in ROS did not increase haematopoiesis.

## Toll signalling promotes haemocyte migration during leg regeneration

We hypothesised that haemocyte migration into regenerating legs would be increased in $Duox^{RNAi}$ nymphs because haemocyte proliferation was not significantly altered (Fig. 5D, Fig. S8C,D). In a previous study, we showed that Toll2-2 promotes plasmatocyte migration into regenerating legs and that the Toll ligands Spz and Spz2 are involved in leg regeneration (Bando et al., 2022).

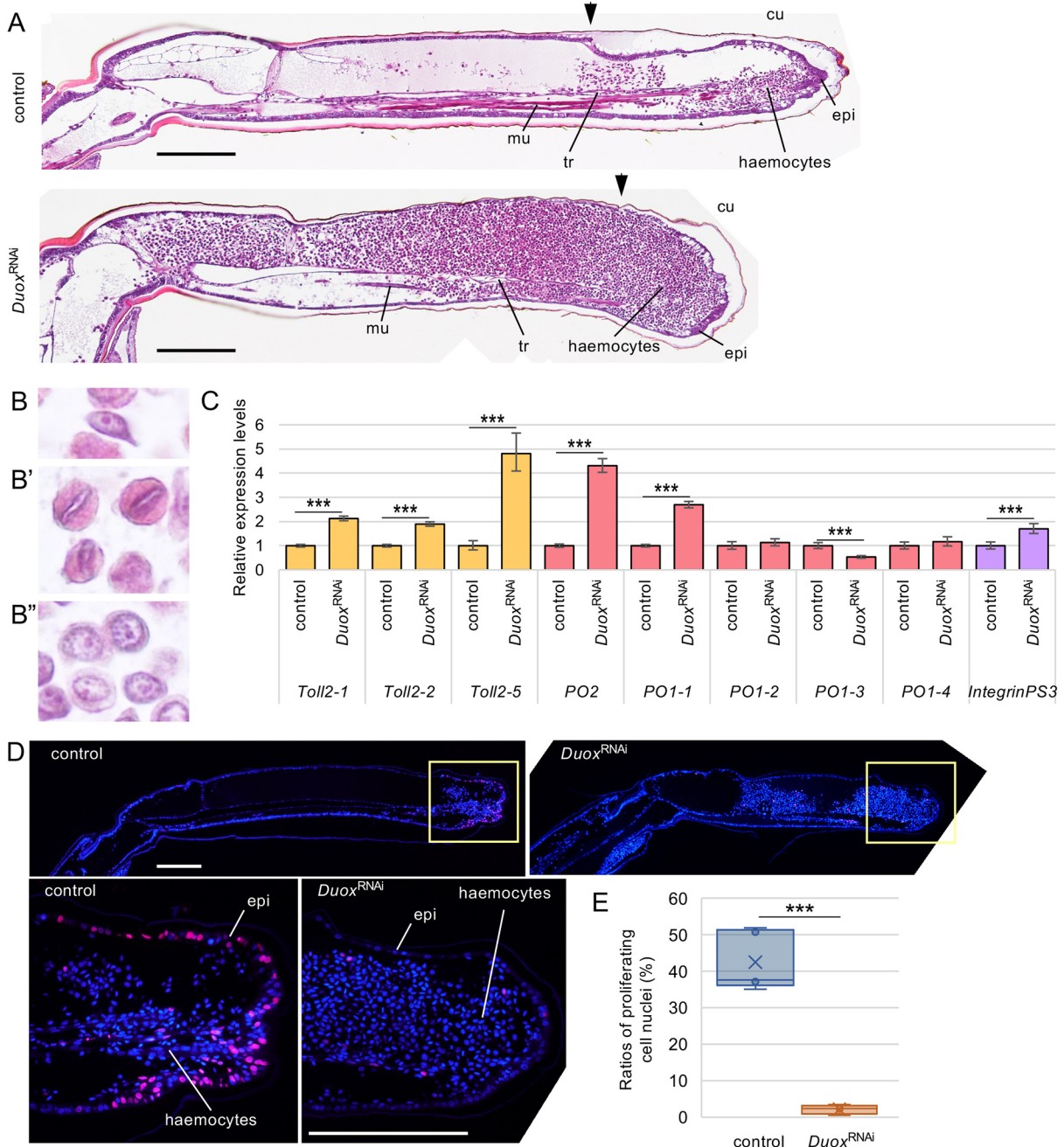

**Fig. 5. Hypertrophy phenotype in _Duox_<sup>RNAi</sup> regenerating legs.** (A) H-E-stained sections of the control and _Duox_<sup>RNAi</sup> regenerating legs at 6 dpa. Arrowheads indicate amputated positions. cu, cuticle; epi, epidermis; mu, muscle; tr, trachea. Images are representative of 17 samples. Scale bars: 200 µm. (B-B″) High-magnification images of the three types of mesenchymal cells – plasmatocytes (B), eosinophilic round-shaped cells with flat nuclei (B′) and basophilic round-shaped cells with round nuclei and a single nucleolus (B″) – detected at increased cell numbers in _Duox_<sup>RNAi</sup> regenerating legs. (C) Relative haemocyte marker gene expression levels in the control and _Duox_<sup>RNAi</sup> regenerating legs at 5 dpa. ***$P<0.001$ (unpaired, two-tailed Student's $t$-test between control and _Duox_<sup>RNAi</sup>). $n=8$. (D) Fluorescence microscopy images of the distribution of proliferating cells at S phase using the EdU incorporation assay in regenerating legs of the control and _Duox_<sup>RNAi</sup> nymphs at 4 dpa. Boxed areas are shown at higher magnification beneath. All nuclei are shown in blue and EdU-incorporating nuclei are in magenta. Images are representative of six samples. Scale bars: 200 µm. (E) The ratios of proliferating cell nuclei of the control and _Duox_<sup>RNAi</sup> regenerating legs are shown in box plots. ***$P<0.001$ (unpaired, two-tailed Student's $t$-test between control and _Duox_<sup>RNAi</sup> regenerating legs). $n=5$.

Therefore, we focused on the role of insect cytokines in haemocyte migration. We determined that the _G. bimaculatus_ genome contains five _spz_ family genes: _spz_, _spz2_, _spz3_, _spz6_ and _Neurotrophin1_ (_NT1_) (Fig. S9A). We quantified the expression changes in _spz_ family genes between control and _Duox_<sup>RNAi</sup> using qPCR (Fig. S10A). During leg regeneration at 5 dpa, _spz_ expression

significantly increased to 156.5% in _Duox_<sup>RNAi</sup> regenerating legs compared with that in control regenerating legs. The expression levels of _spz2_, _spz3_ and _spz6_ decreased slightly to 75.0%, 81.6% and 77.5%, respectively, and _NT1_ decreased markedly to 34.8%.

To examine whether _spz_ family genes are involved in haemocyte migration during leg regeneration, we performed dual RNAi against

*Duox* and each of the *spz* genes and then compared the ratios of the hypertrophy phenotype with *Duox*[RNAi] or single RNAi against each of the *spz* genes (Fig. 6A,B, Fig. S9B). RNAi against *spz* and *spz2* has previously been shown to be effective (Bando et al., 2022) and RNAi against *spz3*, *spz6* and *NT1* significantly decreased the endogenous target RNAs (Fig. S9C). In *Duox*[RNAi] fourth-instar nymphs, 77.1% (*n*=27/35) showed hypertrophy phenotypes, whereas 8.6% (*n*=3/35), 3.9% (*n*=1/26), 9.5% (*n*=2/21), 19.1% (*n*=4/21) and 11.1% (*n*=2/18) of *spz*[RNAi], *spz2*[RNAi], *spz3*[RNAi], *spz6*[RNAi] and *NT1*[RNAi] nymphs showed hypertrophy phenotype,

respectively (Fig. 6B). In the case of dual RNAi nymphs, 34.4% (*n*=11/32), 46.4% (*n*=13/28), 75.0% (*n*=15/20), 75.0% (*n*=15/20) and 94.1% (*n*=16/17) of *Duox/spz*[RNAi], *Duox/spz2*[RNAi], *Duox/spz3*[RNAi], *Duox/spz6*[RNAi] and *Duox/NT1*[RNAi] nymphs showed a hypertrophy phenotype, respectively (Fig. 6B), indicating that the hypertrophy phenotype caused by *Duox*[RNAi] was suppressed in *Duox/spz*[RNAi] and *Duox/spz2*[RNAi], and was unexpectedly enhanced in *Duox/NT1*[RNAi].

To test whether Spz and Spz2 are involved in haemocyte migration, we sectioned the regenerating legs of control, *Duox*[RNAi],

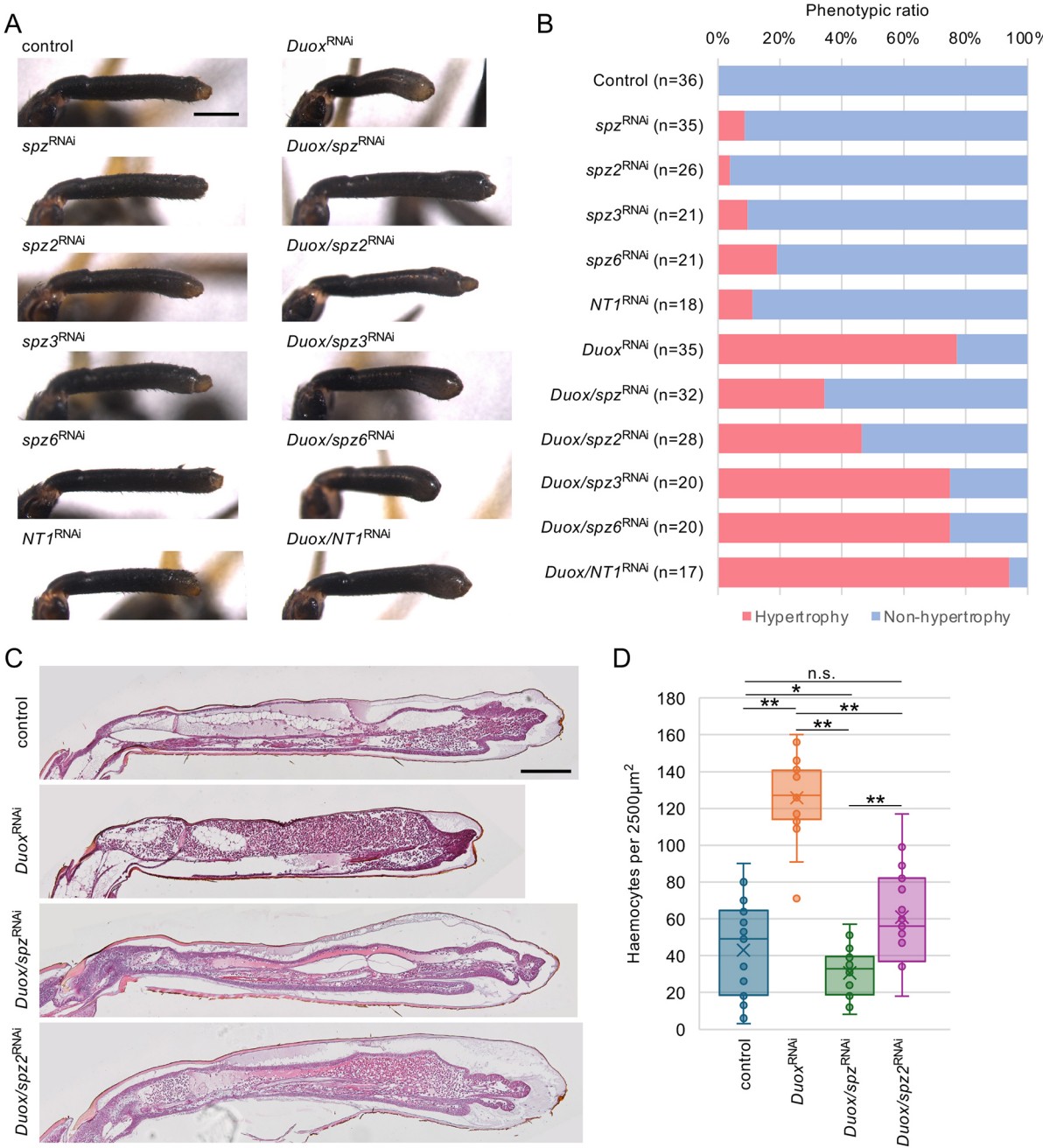

**Fig. 6. Hypertrophy phenotypes and phenotypic rates of dual RNAi against *Duox* and *spz* family genes.** (A) Typical regenerating legs of control, single RNAi and dual RNAi against *Duox* and each of the *spz* family genes at 5 dpa. Scale bar: 500 μm. The number of individuals analysed is shown in Fig. 6B. (B) The percentage of animals exhibiting the hypertrophy phenotype in RNAi nymphs against *Duox* and each of the *spz* family genes at 5 dpa. *n*, RNAi-treated individuals. (C) H-E-stained sections of the control, *Duox*[RNAi], *Duox/spz*[RNAi] and *Duox/spz2*[RNAi] regenerating legs at 6 dpa. Images are representative of four samples. Scale bars: 200 μm. (D) The numbers of haemocytes per 2500 μm² in the regenerating legs of control, *Duox*[RNAi], *Duox/spz*[RNAi] and *Duox/spz2*[RNAi]. *P<0.05, **P<0.01 (Tukey–Kramer test); n.s., not significant. *n*=20.

*Duox/spz*[RNAi] and *Duox/spz2*[RNAi] nymphs (Fig. 6C) and analysed the number of haemocytes in the regenerating legs (Fig. 6D). In the case of *Duox*[RNAi], haemocytes accumulated in the regenerating legs, and the number of haemocytes was significantly increased compared to that in the control. In the case of *Duox/spz*[RNAi] and *Duox/spz2*[RNAi] groups, tarsal segments were regenerated, and haemocytes were significantly decreased compared with that in the *Duox*[RNAi] group (Fig. 6C,D). The number of haemocytes was similar in the control and *Duox/spz2*[RNAi]; however, *Duox/spz*[RNAi] was significantly lower than that in the control or *Duox/spz2*[RNAi] groups (Fig. 6D), indicating that Spz can strongly promote haemocyte migration.

To determine whether the decrease in *NT1* expression (Fig. S10A) was related to the changes in the expression of haemocyte marker genes in *NT1*[RNAi] regenerating legs at 5 dpa, the plasmatocyte marker genes *Toll2-1, Toll2-2* and *Toll2-5*, the oenocytoid marker genes *PO2*, *PO1-1* and *PO1-3*, and the granulocyte marker gene *Integrin PS3*, which were upregulated in *Duox*[RNAi] regenerating legs at 5 dpa (Fig. 5C), were also tested. Among these, *Toll2-1, Toll2-2, Toll2-5*, *PO2* and *PO1-1* were significantly upregulated (Fig. S10B), indicating that the accumulation of plasmatocytes and oenocytoids in *Duox*[RNAi] regenerating legs is partly dependent on a reduction in *NT1* expression (Fig. S10A).

## DISCUSSION

Our previous study revealed that plasmatocytes (insect macrophages) play a role in regeneration (Bando et al., 2022). To clarify the role of the immune response in cricket leg regeneration, we examined the function of ROS, which promote cell proliferation in the regeneration of tissues in aquatic animals. We found that during cricket leg regeneration the level of 4-HNE, a marker of oxidative stress, increased at 2 dpa. RNAi-mediated knockdown of *Gryllus Duox* decreased 4-HNE levels (Fig. 3F) and resulted in delayed wound closure and abnormal scab formation at 2 dpa (Fig. 3B). At 5 dpa, the reduction in ROS production by *Duox*[RNAi] caused hypertrophy in the regenerating legs (Figs 3C-E, 5A); however, epidermal cell proliferation decreased (Fig. 5D), and cuticle formation was abnormally thin (Fig. S5D). During the hypertrophy of *Duox*[RNAi] regenerating legs, haemocytes, including plasmatocytes, oenocytoids and granulocytes, were excessively accumulated (Fig. 5A-C); however, cell proliferation of haemocytes was not increased in *Duox*[RNAi] lymph glands (Fig. S8C,D). We conclude that the excessive accumulation of haemocytes through changes in *spz* family gene expression (Fig. 6) in the thin cuticle caused hypertrophy (Fig. 7).

## ROS is involved in scab formation through cuticle melanisation and stability

When insect tissue is injured, haemolymph leaks. Insect haemolymph contains several types of proteins and cells, and coagulates at the wound surface to form a clot that prevents haemolymph from leaking and inhibits infection. Transglutaminases and phenol oxidases promote haemolymph clotting by cross-linking and melanisation to form scabs (Eleftherianos et al., 2021). Notably, ROS affect transglutaminase activity; thus, *Duox*[RNAi] cricket nymphs exhibit delayed scab formation.

Phenol oxidases, which are produced by crystal cells and lamellocytes in *Drosophila* (Schmid et al., 2019) and oenocytoids

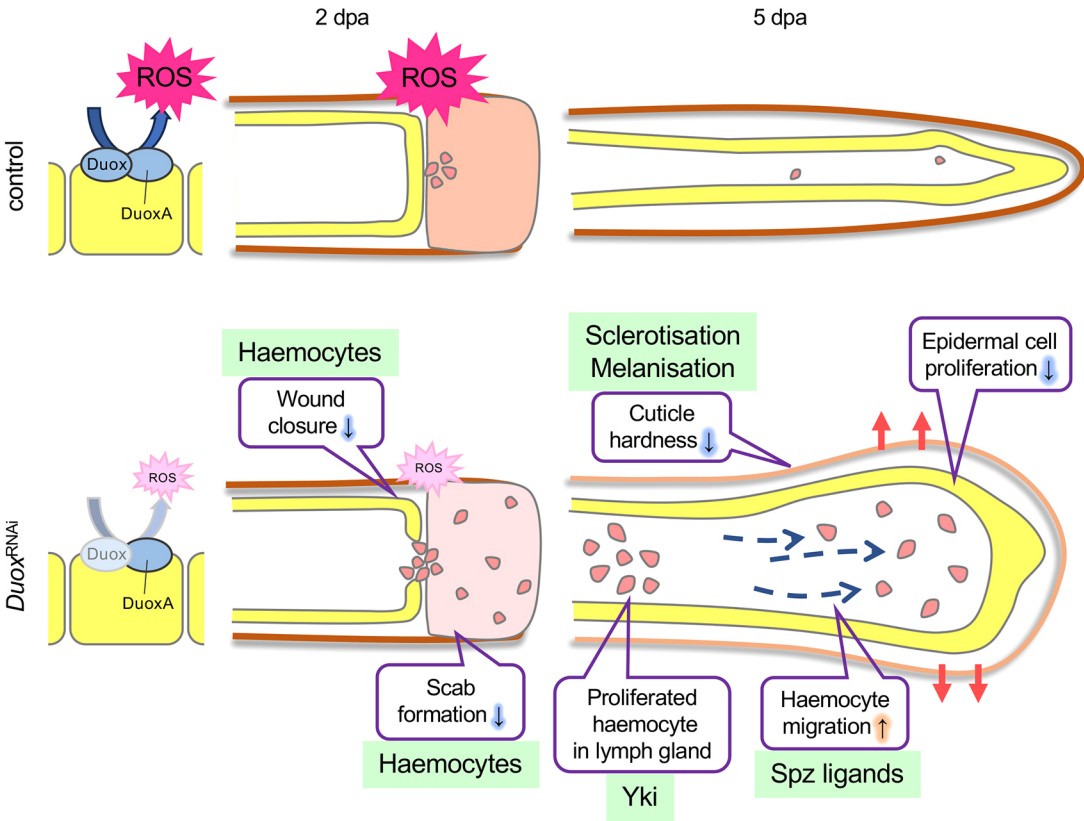

**Fig. 7. Suggested model of *Duox*[RNAi] phenotypes.** Duox and DuoxA are located on the cell membrane and produce ROS. In *Duox*[RNAi] regenerating legs at 2 dpa, scab formation was incomplete, and wound closure was delayed, likely because of lowered plasmatocyte function caused by reduced ROS production. In addition, at 5 dpa, *Duox*[RNAi] nymphs exhibited decreased epidermal cell proliferation, reduced cuticle hardness owing to inadequate sclerotisation and melanisation, and increased haemocyte accumulation in the regenerating legs, possibly because of increased haemocyte migration through Spz and Spz2, and decreased haemocyte reverse migration through NT1.

in other insects (Eleftherianos et al., 2021), promote melanin synthesis. The catechol precursor is required for melanin synthesis and is decreased in *Drosophila Duox* mutants, resulting in defects in melanisation (Thi et al., 2011). In the nematode *Caenorhabditis elegans*, Duox is required for the stabilisation of the cuticular extracellular matrix through cross-linking of di- and tri-tyrosine linkages (Edens et al., 2001). Dityrosine levels in the *Drosophila Duox* mutant are lower than those in the control (Thi et al., 2011). In *Duox*[RNAi] cricket nymphs, we found that the exoskeleton was thinner and paler in colour than that in the control (Fig. S3D), which may have been caused by defects in melanisation and cuticular cross-linking, as observed in *Drosophila* and *C. elegans*.

### ROS produced by Duox is involved in epidermal cell proliferation

Previous studies have reported that ROS bursts occur around wound sites immediately after injury, and ROS production is maintained during tissue regeneration in several regenerative animals (Baddar et al., 2019; Carbonell et al., 2022; Gauron et al., 2013; Jaenen et al., 2021; Khan et al., 2017; Love et al., 2013; Wenger et al., 2014). NADPH oxidase inhibitor-mediated ROS reduction experiments have shown that sustained ROS activates the Wg/Wnt (Love et al., 2013), MEK/ERK (Jaenen et al., 2021), Hippo (Carbonell et al., 2022), JNK (Khan et al., 2017) and other signalling pathways to induce cell proliferation during regeneration. When the lost part of the *Xenopus* tail is regenerated, ROS production decreases (Love et al., 2013). Therefore, the ROS burst is necessary for cell proliferation during tissue regeneration (Serras, 2022).

In crickets, ROS production increased in the epidermis at 2 dpa (Fig. 2), later than that observed in other regenerative animals. Although the increase in ROS production was delayed in crickets, the decrease in cell proliferation in regenerating legs because of ROS reduction (Fig. 5D,E) was consistent with that observed in other regenerative animals (Baddar et al., 2019; Carbonell et al., 2022; Love et al., 2013). In crickets, cell proliferation in regenerating legs is regulated by the Fat/Dachsous, Hippo and Jak/STAT signalling pathways (Bando et al., 2009, 2013). In axolotl tail regeneration, ROS promotes cell proliferation through YAP activation (Carbonell et al., 2022), indicating that Yki, which is an insect homologue of YAP, can be regulated by ROS to promote epidermal cell proliferation during cricket leg regeneration.

In *Duox*[RNAi] nymphs, haemocytes accumulated excessively but did not proliferate in the regenerating legs, causing hypertrophy (Fig. 5A,D); therefore, we speculated that haemocytes proliferate in the haematopoietic tissue. In *Drosophila*, the relationship between haemocyte differentiation and ROS levels is well known (Evans et al., 2003; Owusu-ansah and Banerjee, 2009; Yu et al., 2022). In insect species, haematopoietic stem cells (prohemocytes) as well as subpopulations of plasmatocytes and lamellocytes, are proliferative (Anderl et al., 2016; Asha et al., 2003; Hultmark and Andó, 2022), indicating that proliferating cells in cricket *Duox*[RNAi] lymph glands are both haematopoietic stem cells and plasmatocytes (Fig. S8C). In crickets, RNAi systemically affects the whole body; thus, ROS levels in proliferating cells in the lymph glands should be lower in *Duox*[RNAi] nymphs. The ratio of proliferating cells in the lymph glands of *Duox*[RNAi] nymphs did not change significantly (Figs. S8C,D). In contrast, the hypertrophic phenotype caused by *Duox*[RNAi] was suppressed by dual RNAi of *yki* (Fig. S8A). Yki is a transcriptional co-activator that strongly promotes cell proliferation through the Hippo signalling pathway (Bando et al., 2009), indicating that Yki promotes haemocyte proliferation in the lymph glands rather than through ROS in crickets.

The digestive organs of *Duox*[RNAi] nymphs were also abnormal (Fig. S5B). In *Drosophila*, the gut is highly regenerative following epithelial damage. Damaged intestinal cells release ROS and other stress signals and induce the proliferation of intestinal stem cells through JNK, p38 and Yki activation, as well as cytokine secretion (Jiang and Edger, 2011; Nagai et al., 2021; Zheng et al., 2020). Cricket digestive organ homeostasis can be maintained by a mechanism similar to that of *Drosophila*; however, the regeneration of digestive organs did not occur because the damaged signal through ROS was decreased in *Duox*[RNAi] nymphs. Intestinal homeostasis is impaired by a reduction in ROS levels, which results in gut atrophy.

### Duox-derived ROS promote inflammation resolution

Inflammatory responses have been shown to play an important role in the regeneration of several animals. Immune cells, such as neutrophils and macrophages, quickly infiltrate the wound site in *Xenopus* (Love et al., 2013) and zebrafish (Niethammer et al., 2009) to promote the reconstruction of lost tissues through cell proliferation activation. Infiltrated immune cells subsequently leave the injury site to promote the resolution of the local inflammatory response (Xu et al., 2022). We have previously reported that during cricket leg regeneration plasmatocytes migrate to the wound site and promote leg regeneration through the activation of cell proliferation (Bando et al., 2022). In *Duox*[RNAi] crickets, accumulated haemocytes in the regenerating legs were significantly increased compared to those in the control (Fig. 5), indicating that *Duox*[RNAi] did not cause cessation of inflammation resolution, which promotes haemocyte clearance. We hypothesised that the Spz family cytokines, which are ligands of Toll receptors, would promote haemocyte accumulation as Toll receptor signalling is involved in leg regeneration (Bando et al., 2022). The number of haemocytes accumulated in the regenerating legs caused by *Duox*[RNAi] decreased markedly by dual RNAi against *spz*, but recovered to control levels by dual RNAi against *spz2* (Fig. 6), indicating that *spz* plays a major role in haemocyte migration during leg regeneration.

A previous study showed that inflammation is not resolved when ROS production is decreased by DPI treatment during zebrafish regeneration (Yoo et al., 2012). In crickets, the decrease in ROS caused an increase in haemocytes in the regenerating leg owing to *spz* overexpression (Figs 5 and 6, Fig. S10A), which may disturb the inflammation resolution at 5 dpa.

Insect NT1 is an orthologue of the mammalian brain-derived neurotrophic factor, which plays crucial roles in neuronal cell survival, dendrite and spine elongation, and synapse formation (Lim et al., 2015). Mammalian neurotrophins are also involved in inflammation through eosinophil activity regulation (Nassenstein et al., 2003, 2005). During experimental inflammation of the rat gut, immunoneutralisation of neurotrophins increases the severity of experimental colitis, indicating that neurotrophins also regulate immune cell responses (Reinshagen et al., 2000). Dual RNAi against cricket *NT1* enhanced hypertrophy caused by *Duox*[RNAi] (Fig. 6A,B), and the *NT1* expression decreased in *Duox*[RNAi] regenerating legs (Fig. S10A), indicating that NT1 promotes the resolution of inflammation during leg regeneration. The onset of inflammation, as well as its resolution through Spz family molecules at the appropriate time, appears to be important for leg regeneration.

Previous studies have shown that ROS inhibition causes a decrease in cell proliferation and immune cell migration, resulting in regeneration-defective phenotypes in other aquatic regenerative organisms, such as *Xenopus*, zebrafish and planarians. In contrast, in the cricket, a terrestrial organism with an exoskeleton, ROS bursts

occur at 2 dpa, and appropriate temporal changes in ROS levels are important for scab formation, wound closure, epidermal cell proliferation, haemocyte migration, inflammation resolution, cuticle formation and digestive organ homeostasis. The oxygen environment differs between land and water, with land presenting higher oxidative stress. Mouse foetuses in amniotic fluid regenerate their hearts similarly to aquatic organisms such as newts and zebrafish, but postnatal mice cannot regenerate their hearts (Takeuchi, 2014; Uemasu et al., 2022). The Keap1 protein in terrestrial organisms possesses an amino acid sequence that differs in part from the Keap1 protein in aquatic organisms. Consequently, the Nrf2 degradation activity that responds to oxidative stress in terrestrial organisms is reduced compared to that in aquatic organisms (Yumimoto et al., 2023). Our findings provide insights into a previously unappreciated role for ROS in tissue regeneration in terrestrial organisms. Further comparative studies on the role of ROS are required to support novel approaches to promote tissue regeneration in terrestrial organisms, including humans.

## MATERIALS AND METHODS

### Animals
Nymphs of the two spotted cricket *G. bimaculatus* were reared at a constant temperature of 28°C (Mito and Noji, 2008). Third-instar nymphs were subjected to clodronate liposome (Clo-lipo), RNAi, and DPI treatments to analyse leg regeneration processes (Bando et al., 2017).

### Depletion of plasmatocytes
To deplete plasmatocytes, which are professional phagocytes also known as insect macrophages, we injected 207 nl of Clo-lipo (Macrokiller v100, MKV100, Cosmo Bio) into the abdomens of ice-cold anaesthetised third-instar nymphs using an auto-nanolitre injector (Nanoject II, #3-000-204, Drummond Scientific Company). PBS-liposomes (PBS-lipo) were used as negative controls. Liposome-injected cricket nymphs were maintained separately in insect cages (Bando et al., 2022).

### Preparation of paraffin sections and H-E staining
Regenerating legs were fixed with Bouin's solution (9.1% formaldehyde, 5% acetic acid and 1% picric acid; 023-17361, Fujifilm Wako Pure Chemical Corporation) for 24 h at 4°C. The fixed samples were washed several times with 70% ethanol, dehydrated, and embedded in paraffin. Paraffin sections (5 μm) were prepared with a microtome (RM2145, Leica Microsystems). De-paraffinised sections were stained with Haematoxylin (Mayer's hemalum solution; Merck, 1.09249.0500) for 3 min and Eosin (Eosin Y-solution 0.5% aqueous; Merck 1.09844.1000) for 1 min, or used for immunostaining and observed using a microscope (DM5000 B, Leica Microsystems).

### Cell proliferation assay
Before the fixation of regenerating leg samples, EdU was injected into the abdomen of the third instar nymphs at 2 dpa or fourth instar nymphs at 4 dpa. Injected nymphs were kept at 28°C for 4 h for labelling the S-phase mitotic cell nuclei. The regenerating legs were fixed with Bouin's solution for 24 h at 4°C. Fixed samples were washed, dehydrated, and embedded in paraffin for sectioning. EdU-labelled S-phase cell nuclei were detected using a Click-iT EdU Plus Alexa Fluor 488 Imaging Kit, and total nuclei were detected using Hoechst 33342 (C10637, Thermo Fisher Scientific) using a fluorescent microscope (DM5000 B, Leica Microsystems).

To detect the proliferating cells in lymph glands, EdU-injected nymphs were kept 4 h and fixed in 4% paraformaldehyde in PBT (PBS with 0.05% Tween 20) overnight at 4°C. The fixed nymphs were washed three times with PBT, and the dorsolateral exoskeleton with inner fat bodies was excised using dissecting scissors and tweezers. These exoskeletons were collected in 1.5 ml tubes, and S-phase cell nuclei and total nuclei were detected using a Click-iT EdU Plus Alexa Fluor 488 Imaging Kit and Hoechst 33342 (C10637, Thermo Fisher Scientific) using a fluorescent microscope (DM5000 B, Leica Microsystems).

### ROS detection
De-paraffinised sections of the regenerating legs at several time points were blocked with 5% normal goat serum (NGS; 50062Z, Thermo Fisher Scientific) for 1 h. Blocked sections were then stained with primary antibodies: an anti-4-hydroxy-2-nonenal monoclonal antibody (MHN-020P; JaICA) to detect 4-HNE, or normal mouse IgG for negative control experiments at 1:10 in 5% NGS in PBT overnight at 4°C. Alexa Fluor 488 conjugate anti-mouse IgG antibody (A11029; Thermo Fisher Scientific) was used as the secondary antibody at 1:20 in 5% NGS in PBT for 1 h at room temperature. Hoechst 33342 (H3570, Thermo Fisher Scientific) at 1:500 in 5% NGS in PBT was used with secondary antibody to detect total nuclei. Fluorescence images were captured using a microscope (DM5000B, Leica Microsystems), and fluorescence intensities were measured using the ImageJ software (https://imagej.net/) and are shown as a box plot.

### Cloning of *Gryllus* homologous genes
Nucleotide sequences of *Gryllus* homologues were searched and obtained from the regenerating leg RNA-sequencing database and genome sequences (Bando et al., 2013, 2022; Ylla et al., 2021). Partial gene fragments were amplified using PCR with Ex-Taq or LA-Taq with GC-rich buffer (RR006A or RR02AG, Takara Bio). RNAs extracted from the regenerating legs (RNAqueous-Micro Total RNA Isolation Kit, AM1931, Thermo Fisher Scientific) were reverse-transcribed to cDNAs (SuperScript III First-Strand Synthesis System, 18080051, Thermo Fisher Scientific) using random primers as DNA templates for PCR.

### Phylogenetic analyses and protein domain prediction
The amino acid sequences of Duox, Nox5, DuoxA and the Spz family homologues in insects and crustaceans were identified using the National Centre for Biotechnology Information (NCBI) Protein BLAST (blastp) or tblastn programs (https://blast.ncbi.nlm.nih.gov/Blast.cgi). The CLUSTALW program was used to perform multiple sequence alignments (https://www.genome.jp/tools-bin/clustalw), and a phylogenetic tree was constructed using the FastTree program on the GenomeNet website (https://www.genome.jp/tools-bin/ete).

Protein domains of *Gryllus* homologues of Duox, Nox5, DuoxA, the Spz family, Catalase and Nrf2 were predicted using blastp on NCBI (https://blast.ncbi.nlm.nih.gov/Blast.cgi) and SMART website (http://smart.embl-heidelberg.de/).

### RNAi
The cloned partial fragments of *Gryllus* homologues were used as DNA templates for *in vitro* transcription using the MEGAScript T7 Kit (AMB13345, Thermo Fisher Scientific). Double-stranded RNA (dsRNA) concentration was adjusted to 20 μM. The exogenous gene *DsRed*, encoding a red fluorescent protein from *Discosoma* sp., was used as a negative control. dsRNA (207 nl) was injected into the abdomens of ice-cold anaesthetised third-instar nymphs using an auto-nanolitre injector (Nanoject II), and RNAi-treated nymphs were kept separately in insect cages (Bando et al., 2017). RNAi was performed on ten nymphs, and repeated at least three times to confirm phenotypes.

### qPCR
RNA was extracted from the regenerating legs of eight to ten nymphs using the RNAqueous-Micro Total RNA Isolation Kit and reverse-transcribed into cDNA using the SuperScript III First-Strand Synthesis System. qPCR was performed using the LightCycler Nano System (Roche) and FastStart Essential DNA Green Master Kit (06402712001, Roche). Gene-specific primers were designed using the Edesign website (https://www.dnaform.com/edesign2/). The *Gryllus* β-actin gene was used as an internal control. We ran the qPCR at least three times to generate independent biological replicates.

### ROS synthesis inhibitors treatment
DPI was dissolved in dimethyl sulfoxide (DMSO) at a concentration of 100 μM and stored at −20°C. The 100 μM DPI solution was diluted using distilled water to obtain the required concentrations. DMSO (1%) in distilled water was used as the control.

## Statistical analysis

*P*-values were obtained by unpaired, two-tailed Student's *t*-test, Tukey's test or Tukey–Kramer test. *P*<0.05 was considered significant. For bar graph, bar indicates mean, and error bars indicate s.d. For box plots, horizontal line indicates median, box indicates interquartile range, and whiskers indicate maximum and minimum values of the dataset.

## Acknowledgements

We are grateful to Dr Shiori Ikeda in the Department of Cytology and Histology for technical comments of sectioning, and medical students Nobuaki Fujimori, Shun Manki, Kurumi Saito, Soichiro Shirai and Riko Michishita for their technical assistance with gene function analyses. We thank the staff at the Central Research Laboratory, Okayama University Medical School, for paraffin embedding and recommendations on sectioning.

## Competing interests

The authors declare no competing or financial interests.

## Author contributions

Conceptualization: M.O.-H., T.B., Y.H.; Data curation: M.O.-H., T.B., Y.H.; Formal analysis: M.O.-H., T.B.; Funding acquisition: T.B.; Investigation: M.O.-H., T.B.; Methodology: M.O.-H., T.B., Y.H.; Project administration: T.B.; Resources: M.O.-H., T.B.; Supervision: M.A., H.O.; Validation: M.O.-H., T.B.; Visualization: M.O.-H., T.B.; Writing – original draft: M.O.-H.; Writing – review & editing: M.O.-H., T.B., Y.H., M.A., H.O.

## Funding

This work was supported by Grants-in-Aid for Scientific Research (18K06184, 21K06121 and 24K09412 to T.B.) from the Japan Society for the Promotion of Science. Open Access funding provided by Okayama University. Deposited in PMC for immediate release.

## Data and resource availability

All relevant data and details of resources can be found within the article and its supplementary information.

## The people behind the papers

This article has an associated 'The people behind the papers' interview with some of the authors.

## Peer review history

The peer review history is available online at https://journals.biologists.com/dev/lookup/doi/10.1242/dev.204763.reviewer-comments.pdf

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
