## [Peer Review File · Development (Cambridge, England)]

ROS produced by Dual oxidase regulate cell proliferation and haemocyte migration during leg regeneration in the cricket

Misa Okumura-Hirono, Tetsuya Bando, Yoshimasa Hamada, Motoo Araki and Hideyo Ohuchi
DOI: 10.1242/dev.204763

Editor: Cassandra Extavour

Review timeline

Original submission:	3 March 2025
Editorial decision:	4 June 2025
First revision received:	15 September 2025
Accepted:	16 October 2025

Original submission

First decision letter

MS ID#: dev.204763

MS TITLE: ROS produced by Dual oxidase regulate cell proliferation and haemocyte migration during leg regeneration in the cricket

AUTHORS: Misa Okumura-Hirono; Tetsuya Bando; Yoshimasa Hamada; Motoo Araki; Hideyo Ohuchi

Dear Dr Bando,

Please accept our sincerest apologies for the delay in getting back to you.

I have now received all the referees' reports on the above manuscript, and have reached a decision. The referees' comments are appended below, or you can access them online: please go to:

As you will see, the referees express considerable interest in your work, but have some significant criticisms and recommend a substantial revision of your manuscript before we can consider publication. If you are able to revise the manuscript along the lines suggested, which may involve further experiments, I will be happy receive a revised version of the manuscript. Your revised paper will be re-reviewed by one or more of the original referees, and acceptance of your manuscript will depend on your addressing satisfactorily the reviewers' major concerns. Please also note that Development will normally permit only one round of major revision. If it would be helpful, you are welcome to contact us to discuss your revision in greater detail. Please send us a point-by-point response indicating your plans for addressing the referees' comments, and we will look over this and provide further guidance.

Please attend to all of the reviewers' comments and ensure that you clearly highlight all changes made in the revised manuscript. Please avoid using 'Tracked changes' in Word files as these are lost in PDF conversion. I should be grateful if you would also provide a point-by-point response detailing how you have dealt with the points raised by the reviewers in the 'Response to Reviewers' box. If you do not agree with any of their criticisms or suggestions please explain clearly why this is so.

Reviewer 1

SUMMARY OF THE ADVANCE MADE IN THIS PAPER AND ITS POTENTIAL SIGNIFICANCE TO THE FIELD

The submitted paper explores the role of oxidative stress in regeneration. Specifically, it contributes to the understanding of leg regeneration in *Gryllus bimaculatus* nymphs, which are known to regenerate lost appendages after amputation. Although this species is not a widely used model in Developmental Biology, it has been sporadically employed over recent decades to study regeneration, offering valuable insights from an evolutionary perspective.

The authors build upon previous work showing that depletion of plasmatocytes and RNA interference (RNAi) of Toll receptor ligands Spätzle (Spz) and Spz2 impaired leg regeneration in crickets (Bando et al., 2022). In this manuscript, Okumura-Hirono and colleagues present evidence that RNAi-mediated knockdown of Duox results in delayed wound closure and abnormal scab formation at 2 days post-amputation (dpa). At 5 dpa, reduced ROS production due to Duox knockdown leads to decreased epidermal cell proliferation and an abnormally thin cuticle, likely linked to an overaccumulation of haemocytes and altered expression of spz family genes. This work makes a meaningful contribution to our understanding of the evolution of regeneration mechanisms and supports the idea that oxidative stress plays a regulatory role in tissue regeneration. The paper is potentially interesting and relevant, especially from an evolutionary and comparative biology perspective.

While the data are promising, some results remain preliminary. The manuscript would benefit from further clarification and experimental detail in several areas, as outlined below. Additionally, the text requires some revision for accuracy and clarity.

SUGGESTIONS TO AUTHORS

Major Comments

1. Several statements in the introduction need revision for accuracy. For example:

* The authors write: "ROS are produced in the mitochondria by NADPH oxidase..."

This is inaccurate. While ROS are produced in mitochondria, NADPH oxidase is not a mitochondrial enzyme. Instead, it is typically found in the plasma membrane, phagosomes, and some intracellular vesicles. It generates superoxide (O_2^-) by transferring electrons from NADPH to molecular oxygen, producing $NADP^+$ and H^+ .

Mitochondrial ROS are primarily generated at the electron transport chain, particularly at complexes I and III.

A corrected version might read:

"ROS are produced in mitochondria primarily through leakage of electrons at complexes I and III of the electron transport chain. Separately, NADPH oxidase, a membrane-bound enzyme complex, also produces ROS—specifically superoxide—by transferring electrons from NADPH to molecular oxygen."

* Another sentence states: "Hydrogen peroxide is then catalyzed to hydrogen oxide by catalase and other enzymes."

This should be corrected to:

"Hydrogen peroxide (H_2O_2) is broken down into water (H_2O) and oxygen (O_2) by catalase and other antioxidant enzymes, such as glutathione peroxidase."

In summary, I recommend a careful revision of the manuscript for biochemical accuracy and clarity.

2. The dual RNAi experiments showed that spzRNAi and spz2RNAi suppressed the hypertrophy caused by DuoxRNAi. Could it be possible to test changes in haemocyte migration during leg regeneration? An analysis of the behaviour of cells involved in the hypertrophy would enhance the impact of the paper.

Minor comments

3. Figure 1A, B

Please provide quantitative data regarding the reduction in epidermal proliferation. Is there complete loss of EdU-positive cells in Duox RNAi conditions? Is this consistent across all samples?

4. Anti-4-HNE Antibody

Mention in the text that the anti-4-HNE antibody detects proteins modified by 4-hydroxy-2-nonenal (4-HNE), a product of lipid peroxidation and a marker of oxidative stress.

5. Use of 4-HNE

As 4-HNE is not the most conventional marker for oxidative stress, it would be helpful to note that many standard ROS-detecting dyes cannot be used in fixed samples or in nymph stages, as discussed later in the paper.

6. Figure 2 - Lipid Peroxidation in Muscle at Time 0

The high level of lipid peroxidation in muscle tissue at 0 hpa (far from the wound site) requires clarification. Could the authors comment on the baseline peroxidation in non-amputated legs for comparison?

7. Indicate Haemocytes in Figure 2

Please clearly label haemocytes in the figure and/or legend to help readers interpret the data.

8. Haemocyte Accumulation

The authors state: "Increased ROS production was observed in the haemocytes that accumulate in the clot." However, this accumulation is not clearly visible in the images provided. Consider adding a clearer image or using a haemocyte-specific marker.

9. Fluorescence Quantification Statement

The sentence "The fluorescence intensities of the epidermis were measured using ImageJ and are shown in a box plot" should be moved to the Materials and Methods section.

10. ROS Peak at 48hpa

What is the significance of the observed peak in ROS at 48 hpa? Is there evidence that Duox is more active at this time point?

11. Antioxidant Treatment as a Control

Since RNAi of Duox delayed wound healing, this suggests early ROS production is critical for regeneration. Could the authors test whether exogenous antioxidant treatment (e.g., N-acetyl cysteine) or RNAi of Catalase produces similar effects?

12. Figure 3B - Plasmacyte Accumulation

Please indicate and label the accumulation of plasmacytes in Figure 3B.

13. Gut Atrophy and Mortality

The link between local Duox knockdown in the leg and systemic effects such as gut failure and mortality is not clearly explained. Is there evidence that reduced ROS in the leg disrupts intestinal homeostasis? Further clarification and experimental support are needed.

14. Hypertrophy Characterization

The observed hypertrophy should be further characterized. What cell types are involved? Is this indicative of neoplastic growth, or merely disorganized tissue structure? Could it be driven by secreted factors?

This is a valuable contribution that links oxidative stress to regeneration in an emerging model organism. With revisions and clarifications as suggested above, the paper will be significantly strengthened. I encourage the authors to refine their biochemical explanations, add quantitative data where needed, and expand their interpretation of systemic effects.

15. Key references in the field of ROS and regeneration are missing.

Reviewer 2

SUMMARY OF THE ADVANCE MADE IN THIS PAPER AND ITS POTENTIAL SIGNIFICANCE TO THE FIELD

In this paper, the authors aim to assess the trigger of leg regeneration in the cricket *Gryllus bimaculatus*. They focus on an interesting candidate for this question: the ROS. By combining detection of ROS production and functional assays through many RNAi experiments, they conclude that ROS regulate cell proliferation and haemocyte migration during this process of leg regeneration.

The question is very interesting and the model can provide an interesting perspective on ROS functions during regeneration initiation in Ecdysozoans.

SUGGESTIONS TO AUTHORS

However, I have several comments (majors and minor) on the paper that deal with notably the method used to detect ROS, the interpretation of EdU labelling and the phylogenetic analysis as well as the annotation of the figures.

Major points:

Introduction:

L 50 to 52: the selection of species/ groups cited as examples is not particularly logic. There is a mix of species (hydra) versus groups/lineages for the rest, with a focus on 2 groups of Ecdysozoans which are well known for their relatively poor regenerative abilities. I would recommend to replace hydra by Cnidarians (almost all of them are regenerating very well). "small fish" is not an appropriate term either. A reference is missing to supporting this statement.

L 52 to 55: Regeneration is an extremely complex process that can be virtually divided in 3 common steps and the authors here focuses on the 1st one: the initiation of regeneration. I would thus recommend to cite studies dealing to this step of the process instead of being very vague about the "molecular mechanisms underlying regeneration".

L55 to 58: If links between the immune system and ROS were established during regeneration in some deuterostomian species, this is not what is shown, to my knowledge on the Love et al 2013 paper cited.

ROs are produced in many regenerative context where immune system is barely known, so the production by phagocytes is not always the case. Please rephrase of be more specific.

Results:

Fig 1; for people not familiar with the regeneration of the cricket legs, it would be great to know what is the structure of a non-amputated control and where are the amputation sites in both PBS-lipo and Clo-lipo. Please make annotations to indicate where the wound closure is incomplete in the Clo-lipo in comparison to controls.

L 133 and Fig.1B: Why did you do the Edu labelling on sections and not on the whole leg? Which section is represented (I imagine several sections are produced within a leg). You mentioned that the EdU+ cells are "significantly reduced" upon Clo-lipo treatment of regenerating leg. Did you do counting of the EdU + cells? on sections or on the whole leg? on many animals ? Are the EdU+ cells located uniformly depending on the section considered? Counting must be provided to support the statement. Same comment for the figure 5D.

L135 to 139: the transition is a bit harsh. It is not clear what has been done here and which refers to fig 1c, D. Maybe you should precise that you did qPCR experiments that revealed [...].

L414 and Figure 2: 4-HNE is not a very common tool to detect ROS. The authors must mention why they choose this indirect method and how efficient it is. In particular as the pictures of the Figure 2 are not convincing. The differences of muscle staining by comparing the "control" 0 hpa and the 3 hpa is unexpected and raise question about the method used, as the staining in the cuticles does. Also, the pictures lacks annotations of where eosinophilic clot, haemocytes accumulate etc.

It would be important to have confirmation of the ROS dynamic production by another method.

What about Amplex Ultra Red. This culture medium reporter does not allow to discriminate the location of the ROS+ cells but would confirm the pic at the 2 dpa proposed by the authors.

Fig 3B: same comment as before, please add annotation, (scab, incomplete wound closure, mesenchymal cells, plasmocytes ..)

How can you determine from the Fig 3B, that the "number of mesenchymal cells localized along the trachea and around the muscles are similar between control and Duox RNAi" ? There is no counting performed.

L187: It is unclear why the authors focuses on the digestive system; Is there any other difference visible between the control and Duox RNAi individuals, outside of the digestive system?

Fig 3F: It is difficult to assess which parts of the regenerating legs are illustrated in the panels and as for the figure 2, the labelling are barely convincing. In this figure, the authors quantified the fluorescence both in the epidermis and the mesenchyme and noticed a decrease in Duox RNAi individuals. In the figure 2, in contrast, counting and labelling was only observed in the epidermis. Why there was no signal /measurement in the mesenchyme during the course of leg regeneration?

L 234: the experiment with the DPI drug is interesting but less specific that RNAi (as it targets both Nox and Duox). I would recommend to present it before the RNAi and to perform anti-4HNE on control versus DPI-treated individuals to see if the impact on ROS production is more visible.

L310 to 312: the important information here, are the ratio of EdU+ cells on total cells, the individuals numbers are not necessary to be mentioned as they vary from one individual to another. In addition, interpretation of the data must be done on the ratio only. If the data are not statistically supported there is no trend and here the hypothesis of an increased haematopoiesis in the lymph glands appears not supported. (here and in the discussion part l 417)

Phylogenetic analysis:

The methods used to perform the phylogenetic analysis are missing in the Material and Method part and generally the phylogenetic analysis are not performed properly.

The phylogeny provided in supp figure 1 is very limited in term of species included outside arthropoda (while this do not change the topology of the tree) and there is no node support provided (in contrast to S4).

Supp figure 6 A does not provide any information. There is no outgroup in the phylogeny, that contrains only drosophila sequences in addition to gryllus.

Discussion:

The discussion is a bit long and sometimes redundant (eg L418, L437/438).

I would recommend removing the gut paragraph, a little bit out of focus.

I do not see the point of opposing comparing aquatic versus terrestrial.

Minor comments:

L76 : hydra in italic

L80: in many species mentioned ROS production stops before the blastema formation

L88: in addition to the many pathways mentioned, the importance of apoptosis, which is in many case an intermediated between ROS and proliferation is worth to mention as well.

Legend fig 1: the method to predict domain structures for the Nox5 and duox proteins should be precised in the method part and not in the legend of the figure 1.

Figure 1d: check the statistics, why 3 stars for Nox 5? Not coherent with the text l 138

L 224: The evolutionary history of the Duox MF family is far more complex that initially suspected. See Vullien et al 2025 Mol Biol Evol

Fig 5B: you could identify the pictures by ', ', ' or 1, 2, 3 and refer to them in the text with the proper label (instead of top panel, middle panel, bottom panel).

Fig S5: Please be consistent with color code for EdU (here in green, in other figures Edu is in red)

L 460: the references cited are from 2009 and 2013, so this not very recent. (L 460, recently)

First revision

Author response to reviewers' comments

Reviewer 1: SUMMARY OF THE ADVANCE MADE IN THIS PAPER AND ITS POTENTIAL SIGNIFICANCE TO THE FIELD

The submitted paper explores the role of oxidative stress in regeneration. Specifically, it contributes to the understanding of leg regeneration in *Gryllus bimaculatus* nymphs, which are known to regenerate lost appendages after amputation. Although this species is not a widely used model in Developmental Biology, it has been sporadically employed over recent decades to study regeneration, offering valuable insights from an evolutionary perspective.

The authors build upon previous work showing that depletion of plasmatocytes and RNA interference (RNAi) of Toll receptor ligands Spätzle (Spz) and Spz2 impaired leg regeneration in crickets (Bando et al., 2022). In this manuscript, Okumura-Hirono and colleagues present evidence that RNAi-mediated knockdown of Duox results in delayed wound closure and abnormal scab formation at 2 days post-amputation (dpa). At 5 dpa, reduced ROS production due to Duox knockdown leads to decreased epidermal cell proliferation and an abnormally thin cuticle, likely linked to an overaccumulation of haemocytes and altered expression of spz family genes.

This work makes a meaningful contribution to our understanding of the evolution of regeneration mechanisms and supports the idea that oxidative stress plays a regulatory role in tissue regeneration. The paper is potentially interesting and relevant, especially from an evolutionary and comparative biology perspective.

While the data are promising, some results remain preliminary. The manuscript would benefit from further clarification and experimental detail in several areas, as outlined below. Additionally, the text requires some revision for accuracy and clarity.

Response to Reviewer 1

We express our sincere appreciation to Reviewer 1 for reviewing the manuscript. We agree with all the points raised. We have incorporated the relevant changes, revising the text and figures in accordance with Reviewer 1's comments.

SUGGESTIONS TO AUTHORS

Major Comments

1. Several statements in the introduction need revision for accuracy. For example:

* The authors write: "ROS are produced in the mitochondria by NADPH oxidase..."

This is inaccurate. While ROS are produced in mitochondria, NADPH oxidase is not a mitochondrial enzyme. Instead, it is typically found in the plasma membrane, phagosomes, and some intracellular vesicles. It generates superoxide (O_2^-) by transferring electrons from NADPH to molecular oxygen, producing $NADP^+$ and H^+ .

Mitochondrial ROS are primarily generated at the electron transport chain, particularly at complexes I and III.

A corrected version might read:

"ROS are produced in mitochondria primarily through leakage of electrons at complexes I and III of the electron transport chain. Separately, NADPH oxidase, a membrane-bound enzyme complex, also produces ROS—specifically superoxide—by transferring electrons from NADPH to molecular oxygen."

* Another sentence states: "Hydrogen peroxide is then catalyzed to hydrogen oxide by catalase and other enzymes."

This should be corrected to:

"Hydrogen peroxide (H_2O_2) is broken down into water (H_2O) and oxygen (O_2) by catalase and other antioxidant enzymes, such as glutathione peroxidase."

In summary, I recommend a careful revision of the manuscript for biochemical accuracy and clarity.

Response 1

We have revised the sentences in lines 66-72 as recommended.

2. The dual RNAi experiments showed that *spz*RNAi and *spz2*RNAi suppressed the hypertrophy caused by *Duox*RNAi. Could it be possible to test changes in haemocyte migration during leg regeneration? An analysis of the behaviour of cells involved in the hypertrophy would enhance the impact of the paper.

Response 2

We sectioned and H-E-stained the regenerating legs of the control, *Duox*(RNAi), *Duox/spz*(RNAi), and *Duox/spz2*(RNAi) groups and compared the numbers of haemocytes, as shown in Fig. 6C-D. We found that *spz*(RNAi) or *spz2*(RNAi) significantly lowered the excessive accumulation of haemocytes caused by *Duox*(RNAi) (lines 398-407).

Minor comments

3. Figure 1A, B

Please provide quantitative data regarding the reduction in epidermal proliferation. Is there complete loss of EdU-positive cells in *Duox* RNAi conditions? Is this consistent across all samples?

Response 3

We counted the EdU-positive nuclei and total nuclei and added a graph showing the ratio of EdU-positive nuclei to total nuclei in Fig. 1D, where a marked reduction in the ratio of EdU-positive nuclei was consistently observed in nymphs treated with Clo-lipo (lines 138-140).

4. Anti-4-HNE Antibody

Mention in the text that the anti-4-HNE antibody detects proteins modified by 4-hydroxy-2-nonenal (4-HNE), a product of lipid peroxidation and a marker of oxidative stress.

Response 4

We have included an explanation for 4-HNE in lines 150-152 as suggested.

5. Use of 4-HNE

As 4-HNE is not the most conventional marker for oxidative stress, it would be helpful to note that many standard ROS-detecting dyes cannot be used in fixed samples or in nymph stages, as discussed later in the paper.

Response 5

We have added information about the ROS-detecting reagents that were used but which are not applicable to cricket regenerating legs in lines 152-155.

6. Figure 2 - Lipid Peroxidation in Muscle at Time 0

The high level of lipid peroxidation in muscle tissue at 0 hpa (far from the wound site) requires clarification. Could the authors comment on the baseline peroxidation in non-amputated legs for comparison?

Response 6

We detected 4-HNE in the unamputated leg and observed fluorescent signals in both the epidermis and muscle tissues (Fig. S2A, S2A'). Intense ROS detection in muscle tissues at 0 hpa may be produced by the dominant expression of *Duox* in muscle, as suggested by the qPCR results for *Duox* and *paramyosin* genes (Fig. S3B-C).

Intense ROS detection in the muscle tissue at 0 hpa was reproducible, as confirmed in the other sections (panels on the right). The intense fluorescence in the muscle tissue near the wound site may be the result of muscle tissue shrinkage caused by amputation. We have included an explanation in lines 157-168.

7. Indicate Haemocytes in Figure 2

Please clearly label haemocytes in the figure and/or legend to help readers interpret the data.

Response 7

We have added the labels for haemocytes in Fig. 2B-E.

8. Haemocyte Accumulation

The authors state: "Increased ROS production was observed in the haemocytes that accumulate in the clot." However, this accumulation is not clearly visible in the images provided. Consider adding a clearer image or using a haemocyte-specific marker.

Response 8

We found that plasmatocytes in the accumulated haemocytes produced ROS, as determined by comparison with an India ink-incorporated, non-staining regenerating leg section. This method specifically identifies phagocytes, such as plasmatocytes (Fig. S3A). We have included an explanation in lines 173-180.

9. Fluorescence Quantification Statement

The sentence "The fluorescence intensities of the epidermis were measured using ImageJ and are shown in a box plot" should be moved to the Materials and Methods section.

Response 9

We have moved the sentence to the Materials and Methods section in lines 607-609.

10. ROS Peak at 48hpa

What is the significance of the observed peak in ROS at 48 hpa? Is there evidence that Duox is more active at this time point?

Response 10

We quantified the expression of *Duox* by qPCR, but it was not high at 2 dpa. We also quantified *DuoxA* expression and found that it was the highest at 2 dpa from 3 hpa to 6 dpa. Thus, we concluded that *DuoxA* may activate *Duox* at 2 dpa during leg regeneration (lines 185-195).

11. Antioxidant Treatment as a Control

Since RNAi of *Duox* delayed wound healing, this suggests early ROS production is critical for regeneration. Could the authors test whether exogenous antioxidant treatment (e.g., N-acetyl cysteine) or RNAi of Catalase produces similar effects?

Response 11

We performed RNAi against *Gryllus Catalase* as suggested. We found that *Cat*(RNAi) exhibited abnormal morphology in the regenerated tarsus but did not cause hypertrophy (Fig. S7A-C). We also performed RNAi against *Nrf2*. *Nrf2*(RNAi) did not cause hypertrophy but showed nymphal lethality (Fig. S7D-G). The phenotypes of *Cat*(RNAi) and *Nrf2*(RNAi) were different from *Duox*(RNAi); however, appropriate ROS levels were required for proper leg regeneration (lines 282-293).

12. Figure 3B - Plasmatocyte Accumulation

Please indicate and label the accumulation of plasmatocytes in Figure 3B.

Response 12

We have added the labels for plasmatocytes in Fig. 3B.

13. Gut Atrophy and Mortality

The link between local *Duox* knockdown in the leg and systemic effects such as gut failure and mortality is not clearly explained. Is there evidence that reduced ROS in the leg disrupts intestinal homeostasis? Further clarification and experimental support are needed.

Response13

In the cricket *Gryllus bimaculatus*, RNAi systemically affects the whole body; thus, *Duox*(RNAi) decreases *Duox* expression not only in regenerating legs but also in other organs, including the digestive tract. When we fixed the *Duox*(RNAi) nymphs, we noticed that the *Duox*(RNAi) nymphs did not sink in the fixative solution, although the control nymphs sank. This suggests that air trapped in the digestive tract of the *Duox*(RNAi) nymphs may prevent them from sinking in the fixative solution. We have included a brief explanation in lines 222-224.

14. Hypertrophy Characterization

The observed hypertrophy should be further characterized. What cell types are involved? Is this indicative of neoplastic growth, or merely disorganized tissue structure? Could it be driven by secreted factors?

This is a valuable contribution that links oxidative stress to regeneration in an emerging model organism. With revisions and clarifications as suggested above, the paper will be significantly strengthened. I encourage the authors to refine their biochemical explanations, add quantitative data where needed, and expand their interpretation of systemic effects.

Response 14

We have included an explanation on how hypertrophy occurred by *Duox*(RNAi) in lines 419-429.

15. Key references in the field of ROS and regeneration are missing.

Response 15

We have included references related to ROS and regeneration process.

Reviewer 2: SUMMARY OF THE ADVANCE MADE IN THIS PAPER AND ITS POTENTIAL SIGNIFICANCE TO THE FIELD

In this paper, the authors aim to assess the trigger of leg regeneration in the cricket *Gryllus bimaculatus*. They focus on an interesting candidate for this question: the ROS. By combining detection of ROS production and functional assays through many RNAi experiments, they conclude that ROS regulate cell proliferation and haemocyte migration during this process of leg regeneration. The question is very interesting and the model can provide an interesting perspective on ROS functions during regeneration initiation in Ecdysozoans.

Response to Reviewer 2

We are grateful to Reviewer 2 for the detailed comments on our manuscript. We have performed additional experiments and carefully revised our manuscript in response to Reviewer 2's comments.

SUGGESTIONS TO AUTHORS

However, I have several comments (majors and minor) on the paper that deal with notably the method used to detect ROS, the interpretation of EdU labelling and the phylogenetic analysis as well as the annotation of the figures.

Major points:

Introduction:

L 50 to 52: the selection of species/ groups cited as examples is not particularly logic. There is a mix of species (*hydra*) versus groups/lineages for the rest, with a focus on 2 groups of Ecdysozoans which are well known for their relatively poor regenerative abilities. I would recommend to replace *hydra* by Cnidarians (almost all of them are regenerating very well). "small fish" is not an

appropriate term either. A reference is missing to supporting this statement.

Response

We have replaced “hydra” with “Cnidarians” and “small fish” with “teleost fish”, citing the relevant references, in lines 51-54.

L 52 to 55: Regeneration is an extremely complex process that can be virtually divided in 3 common steps and the authors here focuses on the 1st one: the initiation of regeneration. I would thus recommend to cite studies dealing to this step of the process instead of being very vague about the “molecular mechanisms underlying regeneration”.

Response

We have included an explanation of the three steps involved in tissue regeneration, citing the relevant references, in lines 54-56. In our study, we found that ROS were involved in cell proliferation during regeneration and wound healing in crickets. For clarity, we have included an explanation for ROS production and tissue regeneration (lines 56-65).

L55 to 58: If links between the immune system and ROS were established during regeneration in some deuterostomian species, this is not what is shown, to my knowledge on the Love et al 2013 paper cited.

ROs are produced in many regenerative context where immune system is barely known, so the production by phagocytes is not always the case. Please rephrase of be more specific.

Response

Godwin et al. (2013) first clarified the relationship between regeneration and the immune system. As you mentioned, Love et al. (2013) reported the importance of ROS in tail regeneration. Therefore, we have cited these two references separately in lines 59-62.

Results:

Fig 1; for people not familiar with the regeneration of the cricket legs, it would be great to know what is the structure of a non-amputated control and where are the amputation sites in both PBS-lipo and Clo-lipo. Please make annotations to indicate where the wound closure is incomplete in the Clo-lipo in comparison to controls.

Response

We have added a photograph of a cricket nymph and a schematic illustration of the hind leg of a cricket in Fig. 1A. We also added a label of the wound epidermis (we) in Fig. 1B to show wound closure in the PBS-lipo-treated regenerating leg.

L 133 and Fig.1B: Why did you do the Edu labelling on sections and not on the whole leg? Which section is represented (I imagine several sections are produced within a leg). You mentioned that the EdU+ cells are “significantly reduced” upon Clo-lipo treatment of regenerating leg. Did you do counting of the EdU + cells? on sections or on the whole leg? on many animals ? Are the EdU+ cells located uniformly depending on the section considered? Counting must be provided to support the statement. Same comment for the figure 5D.

Response

In our previous study, we detected EdU-positive nuclei in the whole legs (Bando et al., *Development* 2022). In the present study, we aimed to identify cell types that proliferate. We detected EdU-positive nuclei in the tissue sections. We selected a section of the central part of the leg, including the trachea.

EdU-positive cell nuclei and total nuclei were counted in PBS-lipo- and Clo-lipo-treated regenerating leg sections (Fig. 1D). We also counted the EdU-positive and total nuclei in control and *Duox*(RNAi) regenerating legs, as shown in Fig. 5E.

L135 to 139: the transition is a bit harsh. It is not clear what has been done here and which refers to fig 1c, D. Maybe you should precise that you did qPCR experiments that revealed [...].

Response

We have summarized the results and hypotheses to provide a clear understanding of why we

conducted the experiments described in lines 143-147.

L414 and Figure 2: 4-HNE is not a very common tool to detect ROS. The authors must mention why they choose this indirect method and how efficient it is. In particular as the pictures of the Figure 2 are not convincing. The differences of muscle staining by comparing the "control" 0 hpa and the 3 hpa is unexpected and raise question about the method used, as the staining in the cuticles does. Also, the pictures lacks annotations of where eosinophilic clot, haemocytes accumulate etc. It would be important to have confirmation of the ROS dynamic production by another method. What about Amplex Ultra Red. This culture medium reporter does not allow to discriminate the location of the ROS+ cells but would confirm the pic at the 2 dpa proposed by the authors.

Response

We have included an explanation for the reagents we attempted to use for ROS detection in the cricket regenerating legs in lines 147-155.

Fluorescent signals were observed in the cuticle of the negative control experiments after staining with normal IgG (Fig. S2B, B'). Thus, the cuticle fluorescence was a non-specific signal.

Fluorescent signals in muscle tissues were observed in the non-amputated leg (Fig. S2A, A') but not in the negative control experiment (Fig. S2B, B'), indicating that the fluorescence signals in the muscle tissue were caused by oxidative stress. To clarify whether Duox is expressed in muscle tissue, we compared the temporal changes in the expression of *Duox* and *paramyosin*, a specific marker gene for muscle, using qPCR. *Duox* expression at 0 hpa was high, but low at 3 hpa, 2, 4, and 6 dpa (Fig. S3B). This change in expression was similar to that observed for *paramyosin* (Fig. S3C), suggesting that *Duox* could be expressed in muscle cells. Therefore, the fluorescent signals observed in muscles were the result of specific immunostaining of 4-HNE.

To confirm the highest ROS production at 2 dpa revealed using the anti-4-HNE antibody, we used Amplex UltraRed in the cricket regenerating legs, as suggested by Reviewer 2. Since we do not have established instruments for ELISA, we were unable to detect Amplex UltraRed in ELISA format. Instead, we injected Amplex UltraRed into the cricket nymphs and observed fluorescent signals in regenerating legs to detect ROS via peroxidase activity. Red fluorescence was observed in regenerating legs at 2 dpa (right panel).

We also attempted to establish a method to collect regenerating leg samples at 0 and 3 hpa. However, we were unable to establish this because of the fragile leg tissues; thus, we could not confirm the temporal changes in ROS production using Amplex UltraRed.

AmplexUltraRed detection
in regenerating legs at 2 dpa

Alternatively, we quantified the changes in *DuoxA* expression using qPCR and found that it was the highest at 2 dpa from 3 hpa to 6 dpa, as shown in Fig. S3D. We confirmed that the highest *DuoxA* expression at 2 dpa was associated with the highest ROS production.

Fig 3B: same comment as before, please add annotation, (scab, incomplete wound closure, mesenchymal cells, plasmocytes ..)
How can you determine from the Fig 3B, that the "number of mesenchymal cells localized along the

trachea and around the muscles are similar between control and Duox RNAi" ? There is no counting performed.

Response

We have added labels for wound epithelium (we), plasmatocytes, and incomplete wound closure in Fig. 3B. The quantification of the number of mesenchymal cells along the trachea and muscles was difficult; therefore, we have deleted this sentence.

L187: It is unclear why the authors focuses on the digestive system; Is there any other difference visible between the control and Duox RNAi individuals, outside of the digestive system?

Response

We have included an explanation of the characteristic features of the *Duox*(RNAi) nymphs in fixative solution:

Duox^{RNAi} nymphs did not sink in the fixative solution, whereas control nymphs sank (shown in the right panel). We hypothesised that the *Duox*(RNAi) nymphs may have abnormalities in the digestive tract and dissected them accordingly for observation. We have included an explanation in lines 222-224.

Fig 3F: It is difficult to assess which parts of the regenerating legs are illustrated in the panels and as for the figure 2, the labelling are barely convincing. In this figure, the authors quantified the fluorescence both in the epidermis and the mesenchyme and noticed a decrease in *Duox* RNAi individuals. In the figure 2, in contrast, counting and labelling was only observed in the epidermis. Why there was no signal /measurement in the mesenchyme during the course of leg regeneration?

Response

We have added low-magnification images of the distal regions of the regenerating legs of the control and *Duox*(RNAi) nymphs to show which epidermal tissues were analysed (Fig. 3F).

The fluorescence intensities of both the epidermis and mesenchyme were lower in *Duox*(RNAi) regenerating legs than in the controls. To ensure consistency with the data shown in Fig. 2, the results for the mesenchyme were removed from Fig. 3F, and instead we have focused on demonstrating the results for the epidermis.

In Fig. 2, we have added labels to indicate each tissue in the regenerating leg.

L 234: the experiment with the DPI drug is interesting but less specific than RNAi (as it targets both Nox and Duox). I would recommend to present it before the RNAi and to perform anti-4HNE on control versus DPI-treated individuals to see if the impact on ROS production is more visible.

Response

We established the cricket *Gryllus bimaculatus* as a model insect for tissue regeneration because the RNAi-based analysis of gene function during regeneration can be performed quickly and more easily. Reviewer 2's recommendation has been acknowledged; however, our research focused on analysing tissue regeneration at the genetic level. DPI analysis was performed to confirm the results of gene functional analysis.

L310 to 312: the important information here, are the ratio of EdU+ cells on total cells, the individuals numbers are not necessary to be mentioned as they vary from one individual to another. In addition, interpretation of the data must be done on the ratio only. If the data are not statistically supported there is no trend and here the hypothesis of an increased haematopoiesis in the lymph glands appears not supported. (here and in the discussion part l 417)

Response

As Reviewer 2 mentioned, the number of total nuclei in the lymph glands varied depending on body size. EdU-positive nuclei vary depending on the size of the lymph glands. Thus, we removed the sentences about total nuclei and EdU-positive nuclei and only described their ratios. In statistical analysis, the ratio of EdU-positive nuclei was not significantly changed between the control and *Duox*(RNAi) nymphs. Thus, we have deleted the term “trend” as Reviewer 2 pointed out, in lines 366-368. We have also edited the discussion pertaining to haematopoiesis in the Discussion section (lines 471-486).

Phylogenetic analysis:

The methods used to perform the phylogenetic analysis are missing in the Material and Method part and generally the phylogenetic analysis are not performed properly.

The phylogeny provided in supp figure 1 is very limited in term of species included outside arthropoda (while this do not change the topology of the tree) and there is no node support provided (in contrast to S4).

Supp figure 6 A does not provide any information. There is no outgroup in the phylogeny, that contrains only drosophila sequences in addition to gryllus.

Response

We have included an explanation of the method used for phylogenetic analyses in the Materials and Methods section (lines 621-628). We have also revised the phylogenetic trees for the *Duox* and *Nox5*, *DuoxA*, and *Spatzle* family molecules based on the protein sequences of the insect and arthropod homologues in Fig. S1, S6A, and S9.

Discussion:

The discussion is a bit long and sometimes redundant (eg L418, L437/438).

I would recommend removing the gut paragraph, a little bit out of focus.

I do not see the point of opposing comparing aquatic versus terrestrial.

Response

We have simplified the paragraphs on haematopoiesis and the gut in the Discussion section in lines 471-486 and 487-495, respectively. We have also revised the description of the differences between aquatic and terrestrial species, citing relevant references in lines 537-548.

Minor comments:

L76 : hydra in italic

Response

We have italicized “hydra” in line 82.

L80: in many species mentioned ROS production stops before the blastema formation

Response

As Reviewer 2 mentioned, ROS production decreased after wound healing in zebrafish and axolotl but was sustained during blastema formation in *Xenopus*. In crickets, ROS production increased during wound healing and affected cell proliferation and haemocytes migration after wound healing. We have included citation for references related to both ROS and wound healing, as well as ROS and blastema formation. We have also edited the relevant sentence in lines 84-88.

L88: in addition to the many pathways mentioned, the importance of apoptosis, which is in many case an intermediated between ROS and proliferation is worth to mention as well.

Response

We have added the description “injury-induced ROS promote cell proliferation, which compensates for cell death” in lines 91-93. In our study, we analysed cell proliferation (Fig. 1C, 5D, S8C) but did not perform any analysis on cell death. Thus, we did not provide further detailed explanation for cell death.

Legend fig 1: the method to predict domain structures for the Nox5 and duox proteins should be precised in the method part and not in the legend of the figure 1.

Response

We moved the prediction of domain structures from the Figure legend to the Materials and Methods section (lines 629-632).

Figure 1d: check the statistics, why 3 stars for Nox 5? Not coherent with the text l 138

Response

As Reviewer 2 pointed out, the expression level of *Nox5* in PBS-lipo and Clo-lipo treated regenerating legs was not significantly changed, as revealed by the qPCR results. We have corrected Fig. 1F accordingly.

L 224: The evolutionary history of the Duox MF family is far more complex that initially suspected. See Vullien et al 2025 Mol Biol Evol

Response

We have revised the explanation about DuoxA and cited the recommended reference in lines 259-263. We have also revised the phylogenetic tree of DuoxA in Fig. S6A.

Fig 5B: you could identify the pictures by ', ', "' or 1, 2, 3 and refer to them in the text with the proper label (instead of top panel, middle panel, bottom panel).

Response

We have revised the labels of three types of haemocytes, namely B, B', and B'', in Fig. 5.

Fig S5: Please be consistent with color code for EdU (here in green, in other figures Edu is in red)

Response

We have revised the EdU-positive nuclei to appear magenta in Fig. S8C.

L 460: the references cited are from 2009 and 2013, so this not very recent. (L 460, recently)

Response

We have removed “recently” from line 498, since the references we cited were published a while ago, as Reviewer 2 suggested.

Second decision letter

MS ID#: dev.204763R1

MS TITLE: ROS produced by Dual oxidase regulate cell proliferation and haemocyte migration during leg regeneration in the cricket

AUTHORS: Misa Okumura-Hirono; Tetsuya Bando; Yoshimasa Hamada; Motoo Araki; Hideyo Ohuchi

ARTICLE TYPE: Research Article

Dear Dr Bando,

I am happy to tell you that your manuscript has been accepted for publication in Development, pending our standard publication integrity checks.

Reviewer 1

SUMMARY OF THE ADVANCE MADE IN THIS PAPER AND ITS POTENTIAL SIGNIFICANCE TO THE FIELD

The authors have carefully addressed all my previous comments and concerns. The revisions have improved the manuscript, and I now find it suitable for publication.

Reviewer 2

The authors have addressed all the points raised in my previous review and made all the revisions and clarification suggested.

The new experiments, quantifications, phylogenetic analysis and figure annotations improved the quality of the paper.